# New constraints on Ti diffusion in quartz and the priming of silicic volcanic eruptions

Andreas Audétat [1] ✉, Axel K. Schmitt[2,3], Raphael Njul[1], Megan Saalfeld[4], Anastassia Borisova [5] & Yongjun Lu [6,7]

Titanium diffusion profiles in quartz crystals are widely applied to constrain the duration of magmatic processes. However, experimentally determined Ti diffusion coefficients in quartz diverge by three orders of magnitude. To rectify this problem we derive Ti diffusion coefficients from natural quartz phenocrysts from the 1991 eruption at Mt. Pinatubo, by combining U-Th ages of small (15–40 μm long) zircon inclusions with Ti diffusion profiles measured at nearby growth zone contacts in the same quartz crystals. Application of the obtained data to quartz crystals with Ti-rich rims from thirteen silicic volcanic tuffs worldwide suggests that the magmas erupted years to thousands of years after magma chamber rejuvenation, with the priming time increasing with magma volume and decreasing temperature. Here we show that the time scales involved in the generation of silicic volcanic eruptions are much longer than originally thought.

Volcanic eruptions are commonly interpreted to be triggered by the arrival of new magma batches into subvolcanic magma chambers[1–6]. Whether the eruptions take place immediately after such recharge events, or whether significant time spans elapse in between, is debated. An increasingly popular tool to tackle this question is diffusion chronology on reversely zoned phenocrysts, i.e. crystals that record a last stage of crystal growth in hotter and less evolved magma prior to eruption. Assuming that the contact to this last growth stage was initially sharp, the observed degree of blurring in volcanic crystals can be used to estimate the time between magma recharge and eruption (i.e., "priming time") if the temperature and the corresponding diffusion coefficients are known[7]. A commonly studied mineral for this purpose is quartz because (i) it is a common, alteration-resistant mineral in silicic volcanics, (ii) because its growth zoning can be made visible by cathodoluminescence (CL), (iii) because the CL technique can achieve a high spatial resolution of <1 μm (ref. 8, 9; Methods), and (iv) because the CL intensity of magmatic quartz has been shown to correlate with Ti concentration[3,8,10–12]. However, currently available experimental calibrations of Ti diffusion in quartz diverge by more than three orders of magnitude[13–15], which severely impedes

application of Ti-in-quartz diffusion chronology in understanding volcanic eruptions and hazard prevention.

Experimental determination of Ti diffusion coefficients in quartz has proven to be very difficult because at 800 °C it takes on the order of 100 years to obtain a diffusion profile of only 1 μm in length.

To circumvent this problem, in this work we choose a new approach by deriving diffusion coefficients from natural samples that resided for a known amount of time at known temperatures. For this purpose, 18 small zircon inclusions (10 × 15 μm to 20 × 40 μm in size) within quartz phenocrysts from the 1991 eruption at Mt. Pinatubo are dated via in-situ U-Th disequilibrium age dating (Methods), and nearby Ti diffusion profiles in the quartz are quantified on high-resolution CL images using the grayscale value as a proxy for the Ti content (Fig. 1a). From each zircon inclusion and nearby diffusion profile, a diffusion coefficient is then extracted. In order for this approach to work, the following requirements need to be fulfilled:

(1)  The zircon inclusions should have formed only shortly before their entrapment, which is the reason why only very small grains are chosen for dating. Fulfillment of this requirement is indicated by the fact that the lengths of the nearby diffusion profiles

[1]Bayerisches Geoinstitut, University of Bayreuth, Bayreuth, Germany. [2]Heidelberg University, Heidelberg, Germany. [3]Curtin University, Perth, Australia. [4]Montana State University, Bozeman, MT, USA. [5]Géosciences Environnement Toulouse, University of Toulouse, CNRS, Toulouse, France. [6]Geological Survey of Western Australia, Department of Mines, Industry Regulation and Safety, Perth, Australia. [7]Centre for Exploration Targeting and School of Earth Sciences, University of Western Australia, Crawley, Australia. ✉e-mail: andreas.audetat@uni-bayreuth.de

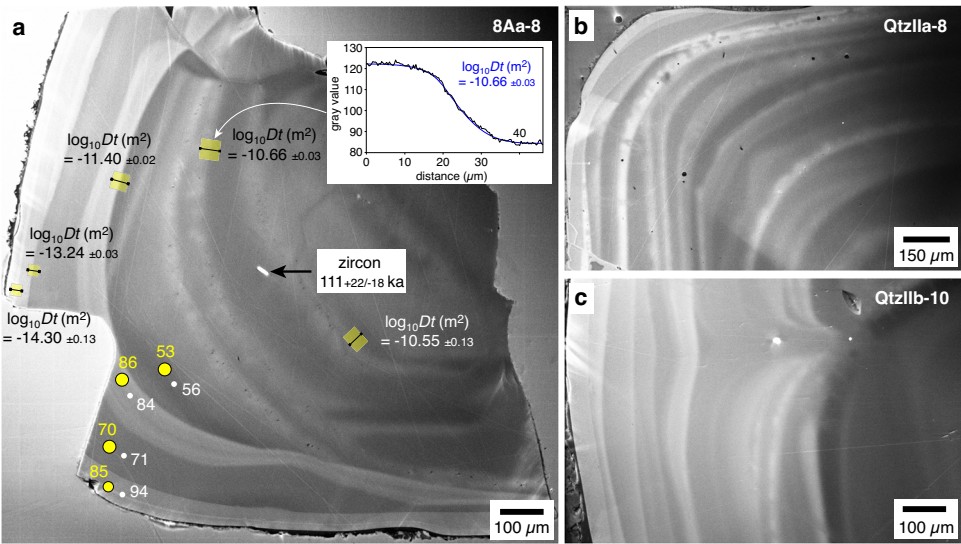

**Fig. 1 | Principle of how Ti diffusion coefficients were derived from natural samples.** The images depict panchromatic (200–900 nm) SEM-CL images taken from three different quartz phenocrysts of Mt. Pinatubo. **a** Sample 8Aa-8, showing a small, exposed zircon inclusion, five measured Ti diffusion profiles (yellow boxes with corresponding $\log_{10}Dt$ values), and locations where the Ti content of the quartz was by EPMA and LA-ICP-MS (white dots and yellow dots, respectively, with the Ti concentrations given in µg/g). From the age of the zircon inclusion and the $\log_{10}Dt$ values of the two nearest diffusion profiles, a diffusion coefficient of $\log_{10}D = -23.2 \pm 0.1$ m²/s was extracted. **b**, **c** Samples QtzIIa-8 and QtzIIb-10, showing growth zone contacts that get increasingly blurred towards the crystal center. Notice that several CL-bright growth zones in Fig. 1a, b were preceded by visible stages of quartz dissolution, as they discordantly cut earlier growth zones.

correlate with zircon age (see below), and that the zircon inclusions are compositionally distinct from zircon microphenocrysts.

(2) The CL images should faithfully monitor the Ti distribution within the quartz crystals. This is demonstrated by spectrally filtered CL-images obtained from selected samples from all occurrences investigated in this study (Methods; Supplementary Data 1), and by the fact that the measured diffusion profiles are 20–200 times longer than the effective spatial resolution achieved during the CL mapping (~0.24 µm; Methods), hence convolution effects are negligible.

(3) The initial Ti concentration step in the measured diffusion profiles must have been relatively sharp, i.e. at least as sharp as the current contacts of the outermost rims. To meet this requirement, diffusion profiles were consistently measured at the inner (older) side of CL-bright growth zones, because these zones seem to have formed in response to heating events (see below) and many of them were preceded by visible stages of quartz resorption (Fig. 1a, b). The assumption of initially relatively sharp contacts is supported by the observation that current contacts are relatively sharp in the outermost growth zones, but become increasingly blurred towards the center of some crystals (Fig. 1a–c), which suggests that the blurring is a consequence of Ti diffusion during magma storage. For the diffusion profiles in the vicinity of dated zircon inclusions, it does not matter whether the initial Ti distribution was an ideal step function or similarly sharp as the current contacts of the outermost rims, as in this specific case the extracted $\log_{10}Dt$ values are affected only on the second decimal place.

(4) The last requirement for our approach to work is that the quartz crystals were stored at a relatively constant temperatures, which is supported by the obtained thermometric data (see below).

## Results and discussion
### New Ti diffusion coefficients

The results obtained from 18 different quartz phenocrysts from Mt. Pinatubo are summarized in Fig. 2. Figure 2a shows that longer diffusion profiles are generally associated with older zircons, whereas Fig. 2b shows that all calculated diffusion coefficients are similar

($\log_{10}D = -23.42 \pm 0.30$ m²/s), independent of zircon age. These two findings concur with the requirements that the zircons formed only shortly before entrapment, and that the quartz crystals were stored at relatively constant temperatures, as demonstrated below.

### Conditions of quartz crystallization and storage

The pressure during quartz crystallization is constrained at: (i) 170–220 MPa by the $H_2O$ content of quartz-hosted melt inclusions (5.5–6.4 wt% $H_2O$[16]; the $CO_2$ content is below 20 µg/g[16]) and petrographic evidence for fluid saturation[17], (ii) $200 \pm 20$ MPa by experimental constraints[18], and (iii) ca. 200 MPa by the major element composition of the residual silicate melt (plotting near the eutectic haplogranite minimum[16]). At the average pressure of $200 \pm 20$ MPa, the $H_2O$-saturated solidus of granitic magmas lies at $683 \pm 6$ °C[19], which provides a lower temperature limit for the quartz crystallization. An upper temperature limit of 760 °C for quartz crystallization is provided by experimental phase relations of the Mt. Pinatubo dacite[18], which is in agreement with the range of 710–750 °C obtained from Fe-Ti-oxide microphenocrysts (Fig. 3). Since Fe-Ti-oxides re-equilibrate relatively quickly[20] and the magma was reheated prior to its eruption, these latter temperatures probably represent maxima. Zircon saturation temperatures constrained from melt inclusions fit well within these temperature limits if calculated based on the zircon saturation model of (ref. 21), model 2 of (ref. 22), or the model of (ref. 23) (Fig. 3). The two other models shown in Fig. 3 provide temperatures that are at least partly below the solidus, and they also give unreasonably high $TiO_2$ activities, for which reason they are discarded. Application of zircon saturation thermometry is justified by the presence of numerous zircon inclusions throughout the quartz phenocrysts (Supplementary Information), and application of the model of (ref. 21) is justified due to the peraluminous nature of the silicate melt.

In the following we prefer the zircon saturation model of (ref. 21) because this model is focused on silicic magmas and relatively low temperatures, and because the temperature range of 690–730 °C obtained with this model fits best with the independent temperature constraints for quartz crystallization in the Mt. Pinatubo dacite. Ti-in-quartz (TitaniQ) pressure calculated for melt inclusion−host quartz

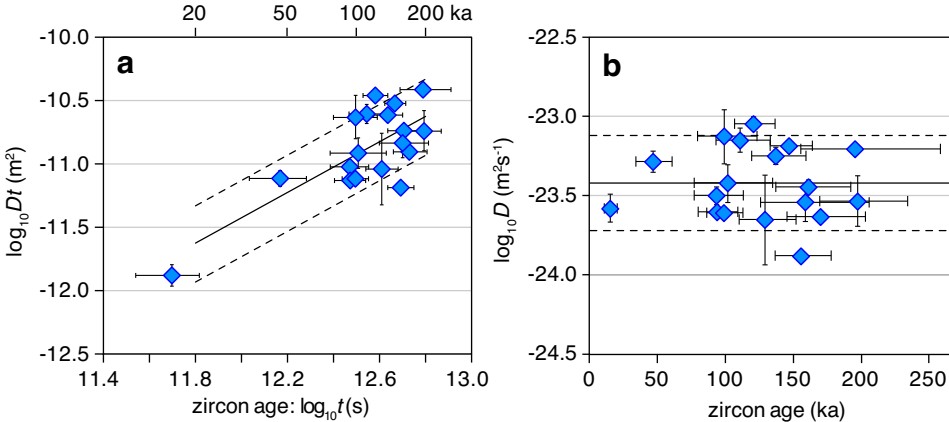

**Fig. 2 | Results obtained from zircon inclusions and corresponding Ti diffusion profiles. a** Sharpness of the measured Ti diffusion profiles as a function of zircon age. The lines show the relationship that would be expected for a constant $\log_{10}D$ value of $-23.42 \pm 0.30$ m²/s. **b** Calculated Ti diffusion coefficients versus zircon age, returning an average $\log_{10}D$ value of $-23.42 \pm 0.30$ m²/s. All error bars denote 1 sigma standard deviation.

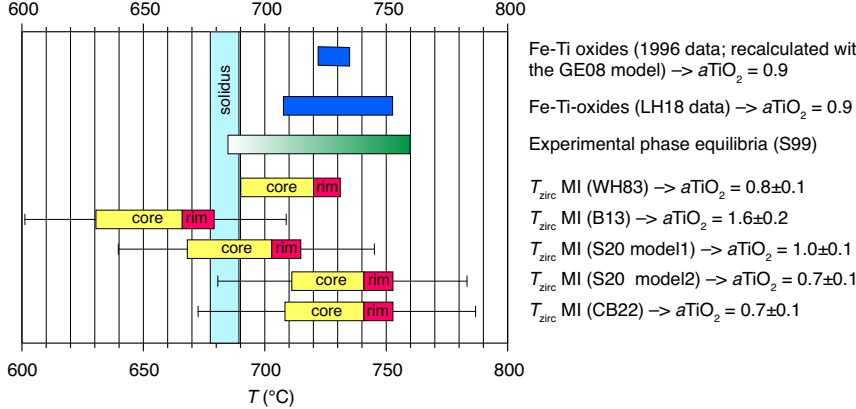

**Fig. 3 | Conditions of quartz crystallization and storage.** The 1996 Fe-Ti-oxide data are from (ref. 2,16,62), and were re-calculated using the model of (ref. 63). The Fe-Ti oxide LH18 data are from (ref. 64) and are also based on the model of (ref. 63). The experimental phase equilibria constraints are from (ref. 18), suggesting onset of quartz crystallization at aqueous fluid saturation at 760 °C. Zircon saturation temperatures and corresponding TiO₂ activities ($a$TiO₂) of melt inclusions hosted in quartz cores versus quartz rims were calculated from their major element composition and Zr contents based on five different models: (1) the original calibration of (ref. 21); (2) the formulation of (ref. 65); (3) model 1 of (ref. 22), which they recommended for metaluminous and peraluminous magmas; (4) model 2 of (ref. 22), which they recommended for alkaline and peralkaline magmas, and (5) the model of (ref. 23). Error bars reflect the standard errors associated with the models. A lower temperature limit for quartz crystallization is defined by the magma solidus at $200 \pm 40$ MPa.

pairs using the approach of (ref. 24), i.e., using the major element and Zr content of the melt inclusions and the Ti content of the corresponding host quartz (using the TitaniQ model of (ref. 25), and the Ti-in-melt solubility model of (ref. 26)), are in excellent agreement with the above pressure constraints, as they all plot within a narrow range of $200 \pm 40$ MPa, except for a single quartz core that appears to have grown at ~300 MPa (Supplementary Information). Importantly, the melt inclusion entrapment temperatures constrained via zircon saturation thermometry show a distinct correlation with the Ti content of the host quartz (Fig. 4), which suggests that the latter reflects dominantly temperature rather than quartz growth rate.

Application of the correlation equation in Fig. 4 to all quartz generations analyzed from Mt. Pinatubo (not only those associated with melt inclusions) returns an extended temperature range of 680–745 °C (Supplementary Information). This range should be representative of the full temperature range that the quartz crystals experienced during their storage because quartz and feldspars are able to dissolve and re-precipitate relatively quickly in response to changes in temperature, hence even if the heating periods are not recorded in the quartz phenocrysts due to quartz dissolution, the subsequent

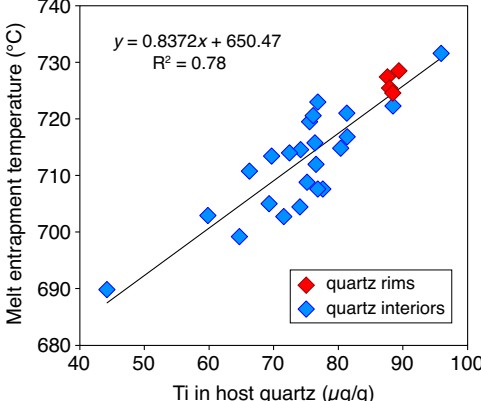

**Fig. 4 | Melt inclusion entrapment temperature versus Ti content of the host quartz.** Melt inclusion entrapment temperatures are based on zircon saturation thermometry using the model of (ref. 21), whereas the Ti content of the corresponding host quartz was determined by LA-ICP-MS.

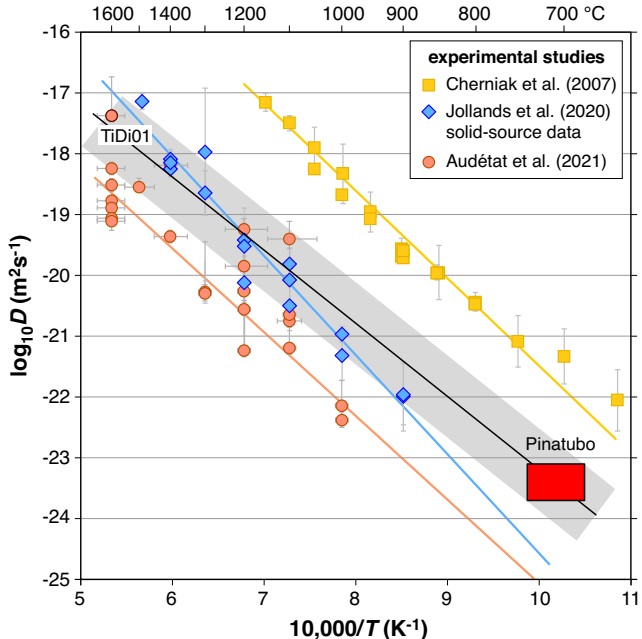

**Fig. 5 | Temperature dependence of Ti diffusion in quartz.** The red box shows the low-T data point derived from the quartz phenocrysts of Mt. Pinatubo, covering the full range of obtained diffusion coefficients and the full uncertainty of the average quartz storage temperature. Error bars of the experimental data were adopted from the literature. At high temperatures we prefer the solid-source experiments of (ref. 14), which agree with experiment TiDi01 of (ref. 15). The solid black line refers to the fitting Eq. (1), with the uncertainty interval (gray array) being determined by the dimensions of the red box.

cooling intervals should be recorded by CL-bright growth zones. If one further considers that magma mush rejuvenation events are characterized by rapid temperature increases that are immediately followed by rapid temperature decreases[27–29], and that no quartz growth occurs during prolonged storage periods at background temperatures, then it is safe to conclude that the *average* storage temperature of the quartz phenocrysts was within 710 ± 30 °C. Notice that it is this time-integrated *average* storage temperature that shaped the diffusion profiles near the zircon inclusions, not the specific temperature at the time of zircon entrapment.

### Temperature dependence of the Ti diffusion

At the reconstructed average storage temperature of 710 ± 30 °C the calculated Ti diffusion coefficients of $\log_{10}D = -23.42 \pm 0.30$ m²/s (Fig. 2b) plot in the middle between the original experimental calibration[13] and the extrapolations of two more recent experimental calibrations[14,15] (Fig. 5). At high temperatures we prefer the solid-source experimental data of (ref. 14) because they are associated with less uncertainty than those of (ref. 15) (Supplementary Information), and because experiment TiDi01 of the latter study, which is the only experiment that was performed on a natural quartz phenocryst similar to those investigated from Mt. Pinatubo (Supplementary Information), fits well with these data. A linear regression through the solid-source data of (ref. 14), forced through the Pinatubo endpoint constrained in the present study, returns the following temperature dependence of the diffusion coefficient (with T given in Kelvin):

$$\log_{10}D(\text{m}^2/\text{s}) = -1.2133*(10{,}000/T - 10.17) - 23.42(\pm0.70) \quad (1)$$

Within the temperature range of 700–800 °C that is most relevant for natural silicic magmas (≥65 wt% $SiO_2$), the diffusion coefficients predicted by this equation are ca. 1.7 log units (50 times) lower

than those of (ref. 13), and 1.5–2.0 log units (30–100 times) higher than the values predicted by the equations in (ref. 14, 15). Any diffusion times within this temperature range that are re-calculated based on these new diffusion coefficients will thus change by the same factors.

### Origin of the Ti-rich quartz rims

The new Ti diffusion coefficients permit estimating the time that elapsed between the beginning of Ti-rich quartz rim growth and the 1991 eruption, based on diffusion profiles measured at core–rim contacts and the estimated temperatures during rim growth. In many quartz phenocrysts the Ti-rich quartz rims grew discordantly over strongly resorbed cores (Supplementary Data 2), and the rims host melt inclusions that are less evolved and return in average 10–30 °C higher zircon saturation temperatures than most melt inclusions trapped in the quartz cores (Supplementary Information). Quartz crystallization pressures reconstructed via Ti-in-quartz (TitaniQ) thermobarometry are 200 ± 40 MPa for both cores and rims (Supplementary Information). These observations suggest that the Ti-rich rims formed in response to magma chamber recharge[3,4,8,24,30] rather than due to decompression[31] or faster growth rates[32]. In most crystals, the partial resorption of Ti-poor quartz cores and subsequent growth of Ti-rich quartz rims represents the most conspicuous feature recorded in the phenocrysts (Supplementary Data 2), suggesting a causal link between the magma chamber recharge and the volcanic eruption. However, renewed growth of quartz after a stage of dissolution requires magma cooling (Supplementary Information), which renders it unlikely that the recharge event recorded by the Ti-rich rims was the *immediate* cause of the eruption. Indeed, the Ti diffusion profiles measured at core–rim contacts return time scales on the order of 30–300 years (Fig. 6), whereas seismic data and the growth of an andesitic dome containing inclusions of freshly quenched olivine basalt suggests that the 1991 eruption was triggered by mafic magma that arrived only days to weeks before the climactic eruption on June 15 (ref. 2, 33, 34). Therefore, at least at Mt. Pinatubo, the Ti-rich rims do not record the recharge event that ultimately triggered the eruption, but one or several previous event(s) that brought the magma chamber into an eruptible state. In that sense, these previous recharge events are still genetically related to the eruption. The fact that total magma residence times recorded by zircon are about three orders of magnitude longer and that no xenoliths of fully solidified dacite were erupted suggests that the Pinatubo dacite magma was stored most of its time at near-solidus conditions in the form of an immobile crystal mush that first needed to be partially re-melted ("primed") in order to become eruptible. This principle has been dubbed "cold magma storage", although exact definitions are diverging[5,35,36].

### Titanium diffusion timescales

The new Ti diffusion coefficients were also used to estimate magma residence times from diffusion profiles measured in the interiors of the quartz phenocrysts. An overview of all obtained data (including those obtained from quartz rims) is provided in Fig. 6, where they are compared with the ages of zircon microphenocrysts, zircon inclusions, and previous eruptions at Mt. Pinatubo. The diffusion ages cluster at 40–400 years, at 3000–20,000 years, and at 40,000–300,000 years. The large number of diffusion ages between 40,000 and 300,000 years is interpreted to reflect quartz growth during establishment of the main magma chamber prior to the ~25 km³ Inararo eruption[37,38] and during subsequent cooling to a crystal mush. The cluster at 3000–20,000 years, which includes also a few core–rim contacts, appears to reflect magmatic activity related to the 10–30 km³ Maraunot to Sacobia eruptions[37,38]. The three data points at ~300 years may be related to the ~5 m³ Buag eruption[37] (notice the time uncertainty introduced by a temperature uncertainty of ±20 °C), whereas all

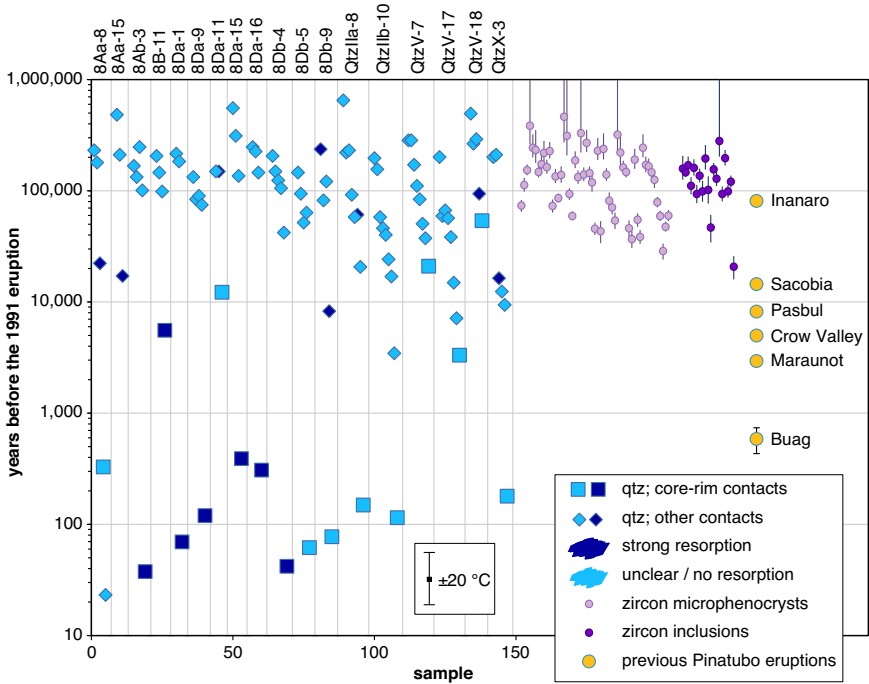

**Fig. 6 | Summary of all age data from Mt. Pinatubo.** Titanium-in-quartz diffusion ages are compared to U-Th zircon ages of zircon inclusions and zircon micro-phenocrysts, and to the ages of previous magma eruptions[37,38]. Error bars of the zircon ages denote 1 sigma standard deviation. Individual quartz phenocrysts are separated by vertical lines. The error bar at the bottom of the graph shows the effect of ±20 °C uncertainty in the estimated magma storage temperature on the calculated diffusion time.

**Table 1 | Overview of studied samples and obtained Ti diffusion time scales**

| Tuff name | Country | Age (Ma) | SiO₂ (wt%) | T (°C)ᵃ | Volume (km³ DRE) | Average residence time (years) | |
|---|---|---|---|---|---|---|---|
| | | | | | | Qtz rims | Qtz cores |
| Sunflower (Lassen) | USA | 0.041 | 68–70 | 850 ± 50 | 3 ± 2 | 30 | 23,000 |
| Pinatubo | PHL | year 1991 | 64–65 | 725 ± 20 | 5.4 ± 0.6 | 100ᵇ | 120,000 |
| Rainbow Mt. | USA | 10.3 | 74–78 | 755 ± 20 | 30 ± 20 | 180 | 12,000 |
| Tunnel Spring | USA | 35.3 | 76 | 710 ± 20 | 50 ± 20 | 120 | 27,000 |
| Bandelier | USA | 1.3 | 74 | 845 ± 20 | 450 ± 50 | 30 | 1300 |
| Amalia | USA | 25.4 | 77 | 725 ± 25 | 750 ± 250 | 2000 | 3800 |
| Cottonwood Wash | USA | 31.1 | 63–68 | 775 ± 25 | 2,000 ± 500 | 780 | 27,000 |
| Hiko | USA | 18.5 | 69–74 | 730 ± 30 | 1,700 ± 300 | 1400 | 7400 |
| Youngest Toba | IDN | 0.075 | 68–77 | 755 ± 25 | 5,300 ± 500 | 330 | 4000 |

*Qtz* quartz, *DRE* dense rock equivalent, *USA* United States of America, *PHL* Philippines, *IDN* Indonesia
ᵃEstimated temperature of quartz rim formation.
ᵇOnly rims that are clearly younger than the Buag eruption.

younger ages are certainly younger than the Buag eruption, and thus are interpreted to record one or several magma recharge event(s) that primed the magma chamber for the ~5 km³ (ref. 39) 1991 eruption, but did not ultimately trigger it. That basaltic magma underplating started considerably before the 1991 eruption has been inferred also from the sulfur evolution recorded in apatite[40]. The bulk of the diffusion age data suggest that the magma chamber below Mt. Pinatubo was established ca. 300,000 years ago and cooled relatively quickly to a crystal-rich mush that became repeatedly re-activated by new batches of ascending, hot magma.

**Ti diffusion time scales obtained from other silicic eruptions**
To be able to compare the results obtained from Mt. Pinatubo with other silicic magma systems, CL images were taken of quartz pheno-crysts from 25 silicic (≥65 wt% SiO₂) tuffs worldwide (Supplementary Information; Supplementary Data 3). Quartz phenocrysts showing Ti-rich rims were found in eight of them (Table 1), but due to the limited number of investigated samples it cannot be ruled out that such phenocrysts occur also in some of the other tuffs, as previous studies have shown that they can be present only in certain tuff layers (typically the last-erupted ones[41,42]).

From the eight samples that contained quartz phenocrysts with Ti-rich rims, time spans between the beginning of rim growth and magma eruption were estimated based on Ti diffusion profiles mea-sured at core–rim contacts, reconstructed pre-eruption magma tem-peratures (Supplementary Information), and corresponding Ti diffusion coefficients according to Eq. (1). If also in these magmas most of the Ti-rich rims are genetically related to the eruption that produced the sampled tuff, then priming times ranging from several years to several thousand years (avg. several hundred years) are indicated, which correlate positively with eruption volume and negatively with magma temperature (Fig. 7, Supplementary Information). Total

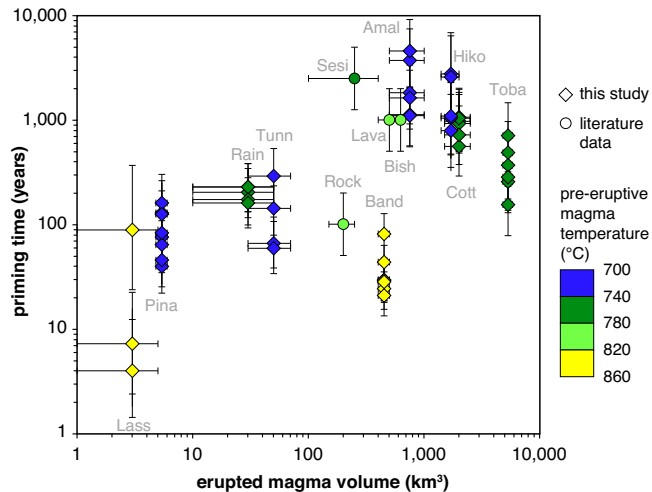

**Fig. 7 | Relation between priming time and erupted magma volume.** Time spans between the beginning of magma chamber rejuvenation and eruption (i.e., priming times), calculated based on Ti diffusion profiles at core–rim contacts of reversely zoned quartz phenocrysts from 13 silicic tuffs worldwide. Data obtained in four previous studies were recalculated based on the diffusion coefficients provided by Eq. 1. The error bars of the priming times denote 1 sigma standard deviation of multiple measurements, whereas the error bars of the erupted magma volumes reflect the range of published estimates. *Lass* Lassen Volcano, *Pina* Mt. Pinatubo dacite, *Rain* Rainbow Mt. Tuff, *Tunn* Tunnel Spring Tuff, *Rock* Rocky Hill ignimbrite[30], *Sesi* Sesia ignimbrite[66], *Lava* Lava Creek Tuff[51], *Bish* Bishop Tuff[1], *Band* Bandelier Tuff, *Cott* Cottonwood wash Tuff, *Amal* Amalia Tuff, *Hiko* Hiko Tuff, Toba Youngest Toba Tuff.

magma residence times obtained from Ti diffusion profiles within the quartz cores range from ~1000 to ~30,000 years (avg. 13,000 years; Table 1; Supplementary Information) and show no correlation to eruption volume or magma temperature, which further supports protracted magma storage at near-solidus conditions in a non-eruptible state. Previously obtained priming times derived from reversely zoned quartz phenocrysts of another four magma systems fit onto this general trend when recalculated based on the diffusion coefficients of Eq. (1), although for the Bishop Tuff also much shorter time scales have been proposed[9].

For some silicic magma systems, independent time constraints have been obtained via Fe-Mg interdiffusion chronometry in zoned orthopyroxene crystals. Corresponding priming times are on the order of decades at Taupo[6,30], and decades to centuries at the Nevado de Toluca volcano in Mexico[43]. These time scales are in broad agreement with those obtained in the present study. Priming times on the order of years to millennia are also in accord with numerical models of silicic crystal mush rejuvenation[44,45]. These models predict that priming times increase exponentially with increasing magma volume, and that they decrease with increasing temperature[45], as suggested by our data in Fig. 7. The higher temperatures of more mafic magmas and the correspondingly lower chance to produce long-lived crystal mushes in the upper crust explains why priming times of more mafic magmas are shorter, typically on the order of days to months[7,46].

In summary, we provide evidence that silicic magma chambers commonly experience decades to millennia of priming before reaching an eruptible state. Therefore, in addition to the classical, relatively short-term eruption forecasting, it may be possible by means of dating of recent geological features (e.g., geothermal fields, minor lava extrusions) and long-term volcano monitoring (e.g., slow uplift, enhanced degassing) to tell whether a given silicic volcano is dormant or currently in a priming, hence potentially dangerous, state.

## Methods

### Sample preparation

Quartz phenocrysts from Mt. Pinatubo, the Amalia Tuff, the Bandelier Tuff, the Tunnel Spring Tuff and from Lassen were handpicked from gently crushed pumice or tuff. Individual quartz phenocrysts were then mounted in epoxy in such a manner that after grinding and polishing with SiC paper and 0.25–3 μm diamond spray they were exposed approximately through the center and roughly parallel to the c-axis of the crystals. The welded tuff samples of the Cottonwood Wash Tuff, the Hiko Tuff, the Rainbow Wash Tuff and the Toba Tuff were first prepared as polished thick sections of 200–500 μm thickness. Parts of these sections were cut out, embedded in epoxy, and then polished to 0.25 μm diamond fineness. In these cases the orientation of the cuts through the quartz crystals could not be controlled. However, care was taken to select only crystals (or parts of crystals) in which the growth zones are oriented approximately perpendicular to the polished surface. All samples were coated with a 12–15 nm thick carbon layer prior to CL imaging or EPMA analysis.

### Zircon U-Th disequilibrium dating of the Pinatubo zircons

Zircon U-Th disequilibrium ages for Mt. Pinatubo were determined by secondary ionization mass spectrometry (SIMS) using the CAMECA ims 1280-HR at Heidelberg University in two analytical sessions targeting (1) individual zircon microphenocrysts embedded in epoxy, and (2) quartz phenocrysts with exposed zircon inclusions, respectively. Primary beam settings ($^{16}O^-$, 30 nA, $30 \times 25$ μm spot) and secondary column tuning follow procedures described in (ref. 47). Twelve different masses used to determine peak intensities and backgrounds were analyzed in a dynamic multi-collection setup with four electron multiplier (EM) detectors with cycles comprising three magnetic field steps that were repeated 30 times per analysis. Gain drift was monitored throughout the analysis sessions by sequential measurement of $^{180}Hf^{16}O$ intensities by the four EM detectors used. A relative sensitivity calibration for $^{232}Th^{16}O$ and $^{238}U^{16}O$ was carried out on AS3 and 91500 reference zircons[48,49] according to the method in (ref. 50). Secular equilibrium zircon references AS3 and Temora 2 (ref. 51) were analyzed in replicate throughout both sessions to ascertain data accuracy. The average $(^{230}Th)/(^{238}U)$ from AS3 ($n = 9$) and Temora 2 ($n = 8$) of $1.017 \pm 0.009$ (1 sigma, MSWD = 0.80) agrees with the expected secular equilibrium value of unity in these materials. For individual zircon microphenocrysts, SIMS spots were placed in close proximity to LA-ICP-MS spots. For zircon inclusions in quartz, targeting of the primary beam on the small (10–20 μm diameter) inclusions was aided by direct ion imaging on a channel plate, where the $^{180}Hf^{16}O$ beam was used for centering the secondary ion beam in the field aperture of the secondary column. Ages for unknowns were calculated as two point model isochrons using measured zircon and published whole-rock data[52], which averaged $(^{230}Th)/(^{232}Th) = 0.879 \pm 0.005$ (1 s.d.) and $(^{238}U)/(^{232}Th) = 0.897 \pm 0.027$ (1 s.d.). All age calculations use the $^{230}Th$ half life of (ref. 53).

### CL imaging and profile quantification

Cathodoluminescence (CL) images were generated with a Zeiss Gemini 1530 field emission gun scanning electron microscope (FEG-SEM) equipped with an ellipsoidal mirror and a photomultiplier detector that detects all light in the range of 200–900 nm. The SEM was operated at 10 kV and an aperture of 120, which corresponds to ~5 nA sample current. CL images were taken at a resolution of $2048 \times 1536$ pixels and a scan rate of 5–10 min/image. For the present samples the use of 10 kV acceleration voltage represents an optimal compromise between high spatial resolution and high signal intensity, providing an effective spatial resolution (full width at half maximum; FWHM) of 0.24 μm (Fig. 8). This value fits with numerical simulations suggesting that 70% of the CL emission in quartz is generated between 20% and

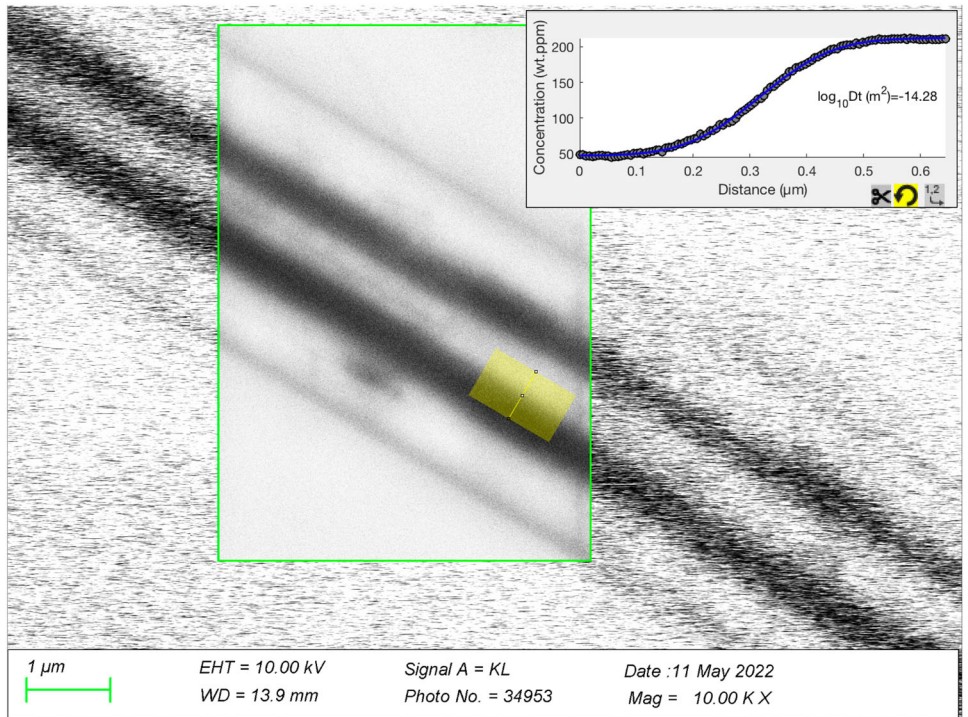

**Fig. 8 | Effective spatial resolution of attained in our CL images.** The image shows a panchromatic CL image obtained from a healed fracture in a quartz phenocryst from the Peach Springs Tuff, Arizona, USA. The fracture is oriented ±perpendicular to the sample surface.

50% of the maximum electron penetration depth, which at 10 kV acceleration voltage is about 1.0 μm[54]. This means that most of the CL emission stems from 0.2–0.5 μm below the surface, where the width of the interaction volume of similar average electron energy is about the same size (i.e., 0.2–0.5 μm). For the profiles measured in the quartz cores, which were all >4 μm, an FWHM of 0.24 μm does not cause any significant convolution effect. For the profiles at the core–rim contact, however, the convolution effect becomes significant and resulted in a lowering of the calculated $log_{10}Dt$ values by up to 0.24 log units.

Initially, all CL images were obtained in panchromatic mode, i.e., by collecting all light in the range of 200–900 nm. However, spectral studies have shown that the CL emission of quartz is produced by various defects, of which only the emission band centered at ~450 nm (~2.7 eV) is related to Ti[10,11,55–57]. The main other emission band of magmatic quartz phenocrysts from tuffs and porphyries is centered at ~650 nm (~1.9 eV) and is related to non-bridging oxygen hole centers (NBOHC). The relative intensities of the two bands vary and can change from Ti-dominated to NBOHC-dominated even within single quartz phenocrysts[10,57] (Fig. 9a, b). It is thus necessary to demonstrate that our CL images taken in panchromatic mode faithfully monitor the Ti distribution in quartz. For this purpose, tests with a 500 nm shortpass filter or a 500 nm longpass filter placed in front of the photomultiplier detector were conducted. The value of 500 nm was chosen because it is in between the ~650 nm band and the ~450 nm band but closer to the latter, such that nearly all of the ~650 nm band is cut off. CL images taken from the same quartz areas once without filter (i.e. in panchromatic mode), once with the 500 nm shortpass filter (i.e., allowing only light with wavelengths <500 nm to pass), and once with the 500 nm longpass filter (i.e., allowing only light with wavelengths >500 nm to pass) are shown in Fig. 9c–g and in Supplementary Data 1. It turns out that in all cases the 500 nm shortpass-filtered images look virtually identical to the panchromatic images and return the same $log_{10}Dt$ values for all analyzed profiles, whereas 500 nm longpass-filtered images look very different. Furthermore, contacts mapped at high

magnification return the same $log_{10}Dt$ values as when they are mapped using the same image acquisition time at lower magnification (Fig. 9c vs d; e vs f), which suggests that the emissions were stable. These observations demonstrate that with our analytical setup and for the samples investigated in this study, panchromatic CL images faithfully monitor the Ti distribution in the quartz.

Grayscale profiles perpendicular to growth zone contacts were extracted with the ImageJ software, using profile widths of 100–300 pixels. Corresponding csv files were then imported into the PACE program[58] and fit with the equation:

$$C(x,t) = C_2 + (C_1 - C_2) \times \frac{1}{2} \times \mathrm{erfc}\left(\frac{x-X}{2\sqrt{Dt}}\right)$$

in which $C(x,t)$ denotes the concentration at position $x$ and time $t$, $X$ the inflection point of the profile, and $D$ the diffusion coefficient ($m^2 s^{-1}$).

### EPMA analysis of quartz

Analyses of Ti and Al concentrations in quartz were performed on a JEOL JXA-8200 microprobe equipped with five spectrometers and TAP, PET, and PETH spectrometer crystals, using 20 kV, 80 nA, and a beam defocused to 10 μm. Following the recipe of (ref. 59), Si was measured 60 s on peak and 30 s on each background (60 s/2 × 30 s), Al was measured three successive times in the same way (60 s/2 × 30 s) on a single TAP crystal, and Ti was measured 180 s/2 × 90 s on two different PET crystals and one PETH crystal. Standardization was performed on corundum (Al), rutile (Ti) and quartz (Si), and a PRZ correction was applied. Test measurements on a natural quartz standard[60] returned average Ti and Al concentrations of 62 ± 3 μg/g and 139 ± 5 μg/g, respectively. Because these averages deviate significantly from the reference values (57 ± 3 μg/g and 154 ± 5 μg/g, respectively), a corresponding correction (in % difference) was applied to all measurements to account for these discrepancies.

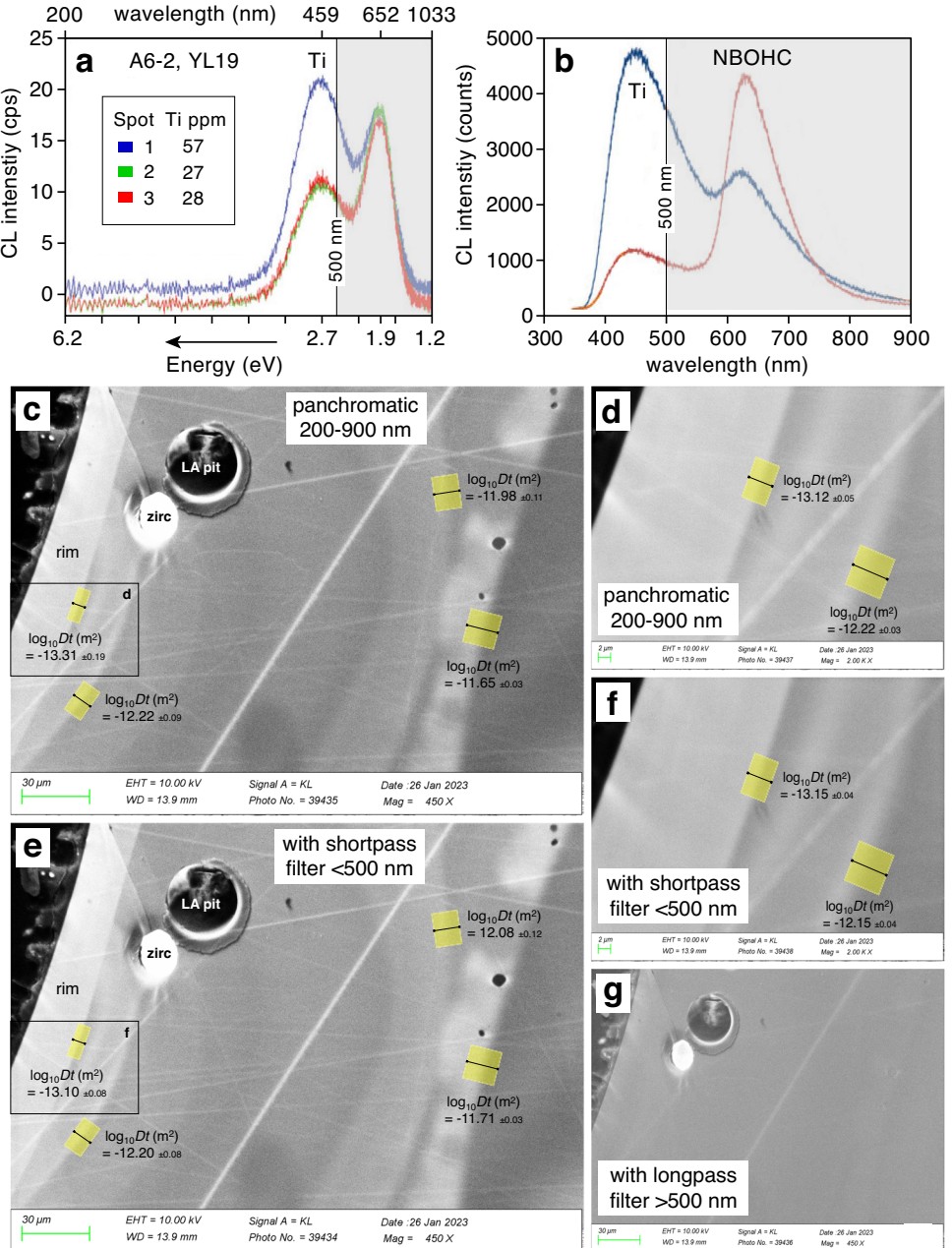

**Fig. 9 | Comparison between panchromatic, 500 nm shortpass-filtered, and 500 nm longpass-filtered CL images. a, b** Published CL spectra of quartz phenocrysts[10,57], showing different proportions of Ti-related emission vs. non-bridging oxygen hole center (NBOHC)-related emission, depending on the analyzed quartz generation (**c, d**) Parts of crystal QtzX-3 mapped at two different magnifications in panchromatic mode. **e, f** The same areas mapped with a 500 nm shortpass filter placed in front of the detector, allowing only Ti-related intensities to pass. Note strong similarity between image pairs (**c, e**) and (**d, f**). **g** The area mapped with a 500 nm longpass filter placed in front of the detector shows no visible zonation.

## LA-ICP-MS analysis of quartz and melt inclusions

LA-ICP-MS analyses of quartz and melt inclusions were performed using a 193 nm ArF Laser (GeolasPro; Coherent, USA) attached to a quadrupole ICP-MS (Elan DRC-e; Perkin Elmer, Canada). The laser fluence at the sample surface was 10–20 J/cm$^2$, and the laser repetition rate was 5–10 Hz. All measurements were performed in a rhombic sample chamber with an internal volume of ~8 cm$^3$, which was flushed with He at a rate of 0.4 l/min, to which 5 ml/min H$_2$ was admixed on the way to the ICP-MS. The ICP-MS system was tuned to a ThO/Th rate of 0.05–0.10% and a rate of doubly-charged Ca ions of 0.15–0.20% according to measurements on NIST SRM 610 glass. Analyzed isotopes comprise $^7$Li, $^{11}$B, $^{23}$Na, $^{25}$Mg, $^{29}$Si, $^{30}$Si, $^{39}$K, $^{43}$Ca, $^{49}$Ti, $^{55}$Mn, $^{57}$Fe, $^{85}$Rb, $^{88}$Sr, $^{89}$Y, $^{90}$Zr, $^{93}$Nb, $^{133}$Cs, $^{137}$Ba, $^{140}$Ce, $^{163}$Dy,

$^{174}$Yb, $^{178}$Hf, $^{187}$W, $^{208}$Pb, $^{209}$Bi, $^{232}$Th and $^{238}$U, using dwell times of 10–50 ms per isotope. External standardization of the major and minor elements in melt inclusions was performed on a rhyolitic obsidian standard[61], whereas NIST SRM 610 was used for the trace elements. Also the Ti concentrations in quartz were externally standardized on NIST SRM610. Internal standardization of the quartz analyses was done using 100 wt% SiO$_2$, whereas for melt inclusions (which were glassy with an exsolved bubble ± a few crystals) the Al$_2$O$_3$ content of a few large, exposed melt inclusions that could be analyzed without co-ablation of quartz host was used. Test measurements on the quartz standard[60] returned average Ti and Al concentrations of $53 \pm 2$ µg/g and $147 \pm 8$ µg/g, respectively, which values agree within uncertainty with the reference values of

$57 \pm 3 \, \mu g/g$ and $154 \pm 5 \, \mu g/g$, respectively. For this reason, no correction was applied in this case.

## Data availability
All data generated in this study are provided in the Tables of Supplementary Data 4.

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

## Acknowledgements

A.A. acknowledges funding by the Deutsche Forschungsgemeinschaft (DFG, German Research Foundation) – 491183248, and by the Open Access Publishing Fund of the University of Bayreuth. Y.L. publishes with the permission of the Executive Director, Geological Survey of Western Australia.

## Author contributions

A.A. designed the study, A.K.S. dated the zircons, R.N. prepared the samples, M.S. and A.B. provided the quartz phenocrysts from Mt. Pinatubo, Y.L. provided the zircon microphenocrysts from Mt. Pinatubo. All authors participated in the data interpretation and contributed to the manuscript writing.

## Funding

## Competing interests

The authors declare no competing interest.
