## [Peer Review File · Nature Communications]

New constraints on Ti diffusion in quartz and the priming of silicic volcanic eruptionsEditorial Note: Parts of this Peer Review File have been redacted as indicated to remove third-party material where no permission to publish could be obtained.

REVIEWER COMMENTS

Reviewer #1 (Remarks to the Author):

In this contribution, Audetat et al. estimate Ti-in-quartz diffusion coefficients at low temperatures (~710 C). They do so using a novel approach, using results from in situ U-Th disequilibrium dating of zircon crystals hosted in quartz crystals and compositional profiles from cathodoluminescence images of the same quartz crystals. They combine their results with those from existing high temperature diffusion experiments to define an equation for Ti diffusion rates in quartz over the temperature range ~700-1600 C. They conclude that Ti-in-quartz diffusivity is intermediate between existing experimental estimates.

They apply their new diffusion coefficients to estimate magma residence and "priming" timescales from cores and rims of quartz crystals from Mt. Pinatubo. They conclude that (a) the quartz crystals existed in a state of cold storage for most of their history, and (b) Ti enriched rims record magma recharge events that do not immediately precede eruption. They apply their new diffusion coefficient to other silicic systems as a comparison and find that quartz crystals in those eruptions also record protracted histories in cold storage and that 10s to 1000s of years of priming (recharging and remelting) the cold system is necessary for a magma to become eruptible.

Audetat et al. have taken a unique approach to address the debate on Ti in quartz diffusivities and the evolution of magmatic systems. They have clearly put in a lot of work and thought about their results. The approach and results have been interesting for me to consider.

I have two overarching concerns that I would appreciate the authors address. I'll note that I have tried to consider the entirety of the dataset; however, there are a lot of parts that go into these results, particularly the temperature estimates, and the authors clearly know and have considered their dataset in detail. So, my (hopefully cogent!) thoughts below may just be points of confusion due to less experience with the dataset and system.

1. Diffusivity as a function of temperature. To define the diffusivity of Ti in quartz as a function of temperature (i.e., equation 1), the authors have chosen a single experimental data point (from a previous study with some of the same authors) at 1600 C and fit a two-point line. While one can certainly do that, I am not convinced it is appropriate here.

Figure 2 highlights experimental results from three different studies for temperatures ranging from 700 to 1600 C. It also shows that a wide range of D values have been estimated at a given temperature within and between these studies. Yet, the authors have selected only a single data point from these three studies, specifically one from the experimental study they conducted. The authors justify their decision to use only this data point from the Audetat et al. (2021) study because it is an experiment on a natural quartz sample and there were some issues with the high temperature experiments they conducted on synthetic quartz. That seems ok. However:

(a) Cherniak et al. (2007) also used some natural samples – at least two experiments in that study are on natural samples – but these were not selected. Why? Including them would certainly change the result.

(b) there are also experiments from another study – Jollands et al. (2020). Do the authors see issues with this data? If not, why not use any of those data points? I'd like to know more about what makes the other experimental data unworthy of use in this case.

In general, if there are issues with any of the experiments that make the authors wary of using them, that would be useful for people to know.

Finally, what rules out the possibility of multiple diffusive mechanisms of Ti in quartz? If there are multiple mechanisms, and if there is some temperature dependence on what mechanism is operating, it seems unwise to create a diffusion relationship based on two points that are endmembers in the temperature range and utilize different methods to constrain D.

Overall, even if the low temperature diffusivities estimated in this study are correct, I am having trouble seeing how the extrapolation here to higher temperatures is appropriate. I know that everyone wants a diffusion coefficient as a function of temperature, but this seems to be oversimplifying a clearly complex problem. In light of this, I also have issues with the authors applying their equation to systems and circumstances with substantially higher temperatures than those determined from the Pinatubo samples.

2. Magma residence times. As the authors note, the temperature at which processes are happening is especially important for diffusion chronometry, and I give them credit for considering the temperature problem thoroughly. I do still have some questions about the temperature estimates and approach (assuming the diffusivities calculated as a function of temperature are appropriate).

First, the text implies that the timescales calculated from quartz interiors are adjusted for temperature using Fig S11; however, in Table S11 it appears that only the rim times were calculated this way. The cores and interiors are instead calculated with the average D value at 710 C (i.e., $-23.36 \text{ m}^2/\text{s}$). So, this is giving a residence time at this temperature. However, the model the authors ultimately come up with for the history of the system negates this, as it implies a non-isothermal history. (e.g., line 167 and line 180-181). Evidence of resorption in many crystals also implies that the temperature got high enough at times to resorb some of the quartz. Thus, assuming a single temperature of 710 for all of the interior boundaries seems inappropriate.

Second, the applicability of zircon saturation temperatures is not clear to me. As I understand it, the logic is: There are zircon crystals and melt inclusions in the quartz, so the melt was zircon saturated when the quartz grew and the zircon saturation temperatures from the melt inclusions reflect the temperature that the quartz grew at. A similar approach was used in a previous study (Audetat 2013) to estimate the temperature of melt inclusion entrapment. In contrast to the current study, however, the 2013 study noted that there was a large abundance of zircon crystals riddling the Bandelier quartz crystals, suggesting the system was zircon saturated throughout the growth history of the quartz. Furthermore, the quartz Ti contents were taken from the same zone where the melt inclusions were located – is this what was done in the current study as well? From the images in the appendix, that doesn't always seem to be the case (e.g., MI3 in 8b-11, MI2 in 8Db-4), in which case the correlation between T_{zirc} and T_{q} in quartz doesn't seem compelling.

In the case where the melt inclusion is located in the same zone as the zircon (and ideally the boundary used for diffusion chronometry), the approach is more convincing. But, as noted above, in most of the images in the supplementary materials, it is not clear that the zones where the zircons are located are at all related to the zones the MIs are in. So, the zircon saturation temperatures from those MIs do not necessarily reflect the temperature at which the zircon inclusions crystallized, let alone the quartz in other zones. Furthermore, the existence of zircon in the quartz does not mean that the quartz grew at the zircon saturation temperature – it just means that the melt was zircon saturated at some point before the quartz crystallized around it. Given the resorption and the dearth of zircon crystals in these crystals (at least compared with the Bandelier samples in the 2013 paper), is it also possible that the system fluctuated between zircon saturated and undersaturated over the history of the system? (It might be worth considering too that the zircon crystals themselves had to grow – taking the maximum size of your crystals (40 μm diameter) and the fastest existing estimates for zircon growth rates (10-15 m/s) would imply that these crystals had to grow for nearly a millennium before being captured in the quartz crystal – how much could temperature have changed in that time?).

Also, as noted in my detailed comment about Fig 3, it is interesting that the zircon inclusions all seem to have ages that are within a constrained range of time, both relative to zircon microphenocrysts and the history of the system as a whole (i.e., there are older and younger microphenocrysts, and there are no zircon grains – inclusions or microphenocrysts – younger than the Sacobia eruption). What, if anything, does this mean about the zircon inclusions being representative of the time when the quartz grew?

This is all to say that I am not sure that zircon saturation temperatures are necessarily providing useful information for determining the temperature of quartz crystallization related to the

boundaries used for diffusion chronometry in the current study. The fact that the authors see such a wide range of T_{zirc} – up to 20 °C at 70-80 ppm Ti in quartz (Fig S11) – at a given Ti in quartz value also makes the use of this correlation for T_{qtz} somewhat unsatisfying. Again, my concerns here may just reflect the fact that I don't have as much experience as the authors with the intricacies of this dataset.

I'm also curious about the calculation of the zircon saturation temperatures - it appears that the W&H zircon saturation temperatures were selected because of their similarity to Fe-Ti oxide temperatures and a_{TiO_2} ; however, as the authors note, the Fe-Ti oxides can be unreliable, and processes like eruptive decompression can negate their robustness (even if they pass the Bacon & Hirschmann test – see your ref Hau 2021). The T_{zirc} results from the Shao 2020 model 2 are significantly higher (but also within the range of the Fe-Ti oxide temperatures) – are these being discounted based on magma composition? (I realize that the Shao et al. (2020) study suggests Model 1 is better for metaluminous to peraluminous systems, but in this study the fact that they give temperatures at the solidus makes it seem less reliable.)

Third, the existence of compelling resorption textures suggests that the temperature was high enough to resorb quartz at one or more times in the history of the quartz crystals (as you note, for example, in grains 8Aa-8 and 8Aa-15). Even if you do not have timescales and temperatures from these boundaries, it seems hard to interpret these textures as not reflecting some kind of resorption event. Perhaps there are other changes to the system other than temperature, but you at least rule these out for many of the boundaries. So, assuming that all of these boundaries are related to thermal fluctuations (arguable if that's fair), it is evidence of multiple temperature fluctuations. How, then, can the timescales calculated for the interior boundaries not be maxima? Even if the system cooled down and began crystallizing quartz again after a high temperature excursion, the time spent at high temperature would impact the diffusivity. Unfortunately, there is no way that I can see here that you could constrain how long the system was at these higher temperatures, or even what those temperatures were, so quantifying how much diffusion would have taken place at the higher temperatures is challenging. But, it seems important to think about.

A few additional comments and questions:

Line 156: missing end parenthesis

Line 174-176: the implication in this sentence is that the system went through multiple periods of recharge. How did the system vacillate between warm and cold but not produce evidence of resorption? Maybe they were relatively short-lived events or didn't reach temperatures where the quartz was resorbing, but this sentence at least implies that there were temperature fluctuations that would impact the diffusion coefficients.

Figure 3: Using the same color for "unclear" and "no resorption" in this plot is a bit misleading. Looking at the images for some of the grains, (e.g., 8Da-1), the contact where the interior diffusion profiles have been taken does not at all appear to be normal, and that deserves delineation on this plot. How do you explain the younger "other" contact in 8Aa-8? Why are all the of the zircon inclusions from a fairly tightly constrained age range? Geochemically the included zircons look significantly different from the microphenocrysts as well. Is this related to a size bias, since you are trying to select the smallest included zircon crystals? Or does it mean something about the other crystals (e.g., older grains are xenocrysts)? Similarly, why are there so few younger grains? In general, why are all the analyzed zircons seemingly Inanaro to a bit younger in age? What does this mean about your assumptions of zircon saturation? If you compare similarly sized microphenocrysts and inclusions, do you estimate similar ages and/or temperatures?

Table 1: The acronyms for the country should be in the caption

Lines 442-446: Why were the standards not reproducible?

Line 463: what does 'mostly glassy' mean? Were they devitrified or is it just that they sometimes also contained a bubble and crystals?

Reviewer #2 (Remarks to the Author):

In this manuscript, the authors use Ti diffusion profiles in natural quartz to extract diffusion coefficients, with absolute time scales constrained using zircon ages. The concept is novel and solid, and the data are high quality.

There is a major issue that has not been addressed satisfactorily, which is the nature of the initial condition prior to Ti diffusion. The inherent assumption in the model is that the Ti profile is a perfect step function, but it is not clear how this assumption is backed up by the data/textural observations etc. What is the evidence that growth is not accompanied by some progressive change in the composition of the growth medium, leading to a gradual change in the quartz composition? Then, the initial condition is not a step function, and diffusion only smooths out some pre-existing gradient. The result would then be that the diffusion coefficients determined are maxima.

As a result, the new data point determined from natural quartz is not a single point on an Arrhenius plot, and the extrapolation between an experimental datum and this one is not universally applicable. This issue then propagates into all of the other time scales.

Therefore, whilst this study provides high quality data, I would not recommend publication in this form. As it stands, the manuscript will further confuse the Ti in quartz debate. The authors should either provide clear and solid evidence that the initial condition prior to diffusion in the natural system was a step function, or alternatively reframe the discussion around the new data simply providing a maximum diffusion coefficient for Ti, and that the actual diffusion coefficient must be somewhere below this value. This would obviously be a major revision.

Reviewer #3 (Remarks to the Author):

Review of NCOMMS-22-45043-T: New constraints on Ti diffusion in quartz and the priming of silicic volcanic eruptions

Dear Dr. Andreas Audétat,

It was a pleasure to review your manuscript. As you know, I agree that Ti in quartz diffusion is one of the great problems facing petrologists who utilize the technique for magmatic time scales. I found the writing to be thoughtful and well researched, mostly. For me, there are two main flaws that crop out during the review. I think that this is an important enough issue to suggest that these two flaws should be addressed before this can be published. I would hate to continue to add controversy to this problem, when you have a real chance to provide some clarity.

Below I'll outline my two main issues, and I hope you find the comments helpful.

Best,

Dr. Joe Boro

Senior Staff Scientist and EPMA Lab Manager

Sandia National Labs, USA

1. You correlate greyscale CL to Ti concentration and use it as an exact proxy. This is incorrect to do, especially as it appears the data were collected on an SEM and not a microprobe (explained further below). The reasons to avoid this are vast, and although you provide a plethora of researchers who choose to do this (and you've done this before), they are also incorrect, so I'm afraid your application of the technique here only muddies the waters further. Boro et al. 2021 (Am. Min.) as well as other authors, give a large list of why not to do this, and I can summarize here for you:
 - a. Boro et al. (2021) find Ti profiles measured by EPMA and overlapping greyscale CL profiles do not match in most cases, often the grey-scale CL showing a steeper or more diffuse gradient. Matthews et al. (2012), their Figure 7a, also show this mismatch (even though they ignore it's significance in their data interpretation).
 - b. While spectroscopic measurements of quartz CL emissions at specific wavelengths, such as ~454 nm, are very well correlated with Ti contents (MacRae et al. 2013, 2018), grayscale CL is not a reliable proxy for Ti concentration.
 - c. Also, as described by Leeman et al. (2012) and MacRae et al. (2018), grayscale intensities include not only CL from Ti dopants but also from the intrinsic CL produced by quartz, by crystallographic defects, by aluminum dopants, and by non-bridging oxygen holes in the crystal structure
 - d. For many panchromatic CL detectors, the collection optics accept light from a wide variety of angles, and therefore light from adjacent areas, which are still emitting from the initial electron bombardment, or light transmitted through the sample and scattered off microcracks in the sample, which may be included in a

CL measurement that is supposed to represent only the CL emitted from an individual pixel (MacRae et al. 2013).

- e. This should be especially avoided in SEMs where there are not optical focusing capabilities. Often times stray photons can bounce off the interior of the chamber and add noise to measurements, artificially adding counts to adjacent pixels during the raster or line scan, and creating artificially diffuse profiles.
 - i. One exercise you could do, is to use extremely low kV (2-5 kV) with fast scan rates, to help with any over excitement of the bonds.
- f. This issue isn't as much of a problem outside of the growth boundaries and is likely why you see CL correlating with Ti in the spot analyses presented in your supplemental data. See Matthews et al., their Figure 7a especially demonstrates the mismatch at the diffusion boundary, but the correlation outside the boundary. I think this has something to do with the intrinsic defects which get included during rapid growth of the quartz, which are strong CL producers. Boro et al. (2021) also sees good correlation outside the growth boundary. For what it's worth, later on we did collect spectral CL at 452 nm, and were able to match that pretty well with the EPMA Ti data, so if you have access to a spectral CL system, I suggest using that as it is better proven to act as a proxy for [Ti]. I realize collecting trace quant data for Ti in quartz at these scales is difficult, but methods do exist (nanoSIMS, oblique EPMA profiles, etc.), or at worst, get some spectral CL data.
- g. Lastly, there is no proof that Ti chemical profiles are thought to start as step functions, which is fundamental to this paper to carry any meaning. A study which examines [Ti] at $T=0$, say in growth experiments would be much more impactful and helpful in understanding the diffusivity discrepancies.

To summarize, you are assuming greyscale CL is a direct proxy for Ti, which we know isn't true, especially at diffusion boundaries, and then using unknown $T=0$ diffusion gradients to extract a diffusivity coefficient. I hope you can see why this raises red flags in my mind.

- 2. You assume that the zircon inclusions found in the quartz grew and formed at the same time as the rims of the quartz. I find this assumption to also be simplified and ignoring some simple thermodynamics about zircon saturation. For example, if Ti activity is increased due to elevated temperature in a system to increase the Ti in a quartz rim, zircon wouldn't be crystalizing, and possibly resorbing. As I see in most of your images, the zircons are lacking idiomorphism and are in most cases rounded, looking to be resorbed, suggesting they existed in the system for some time prior to being incorporated in the quartz rim at a time when they themselves were not in equilibrium.

RESPONSE TO REVIEWER COMMENTS

(comments shown in black; responses shown in blue)

Reviewer #1 (Remarks to the Author):

In this contribution, Audetat et al. estimate Ti-in-quartz diffusion coefficients at low temperatures (~710 C). They do so using a novel approach, using results from in situ U-Th disequilibrium dating of zircon crystals hosted in quartz crystals and compositional profiles from cathodoluminescence images of the same quartz crystals. They combine their results with those from existing high temperature diffusion experiments to define an equation for Ti diffusion rates in quartz over the temperature range ~700-1600 C. They conclude that Ti-in-quartz diffusivity is intermediate between existing experimental estimates.

They apply their new diffusion coefficients to estimate magma residence and “priming” timescales from cores and rims of quartz crystals from Mt. Pinatubo. They conclude that (a) the quartz crystals existed in a state of cold storage for most of their history, and (b) Ti enriched rims record magma recharge events that do not immediately precede eruption. They apply their new diffusion coefficient to other silicic systems as a comparison and find that quartz crystals in those eruptions also record protracted histories in cold storage and that 10s to 1000s of years of priming (recharging and remelting) the cold system is necessary for a magma to become eruptible.

Audetat et al. have taken a unique approach to address the debate on Ti in quartz diffusivities and the evolution of magmatic systems. They have clearly put in a lot of work and thought about their results. The approach and results have been interesting for me to consider.

I have two overarching concerns that I would appreciate the authors address. I'll note that I have tried to consider the entirety of the dataset; however, there are a lot of parts that go into these results, particularly the temperature estimates, and the authors clearly know and have considered their dataset in detail. So, my (hopefully cogent!) thoughts below may just be points of confusion due to less experience with the dataset and system.

1. Diffusivity as a function of temperature. To define the diffusivity of Ti in quartz as a function of temperature (i.e., equation 1), the authors have chosen a single experimental data point (from a previous study with some of the same authors) at 1600 C and fit a two-point line. While one can certainly do that, I am not convinced it is appropriate here.

As explained on lines 112-115 of the original manuscript, this point was chosen because this experiment was performed on a magmatic quartz phenocryst similar to the samples investigated in the present study, whereas all other experiments were performed on hydrothermally grown quartz. However, virtually the same temperature dependence would be obtained if the solid-source data of Jollands et al. (2020) are used as the high-T endmember. Corresponding text was added to the main text and the Supplementary Information.

Figure 2 highlights experimental results from three different studies for temperatures ranging from 700 to 1600 C. It also shows that a wide range of D values have been estimated at a given temperature within and between these studies. Yet, the authors have selected only a single data point from these three studies, specifically one from the experimental study they conducted. The authors justify their decision to use only this data point from the Audetat et al. (2021) study because it is an experiment on a natural quartz sample and there were some issues with the high temperature experiments they conducted on synthetic quartz. That seems ok. However:

(a) Cherniak et al. (2007) also used some natural samples – at least two experiments in that study are on natural samples – but these were not selected. Why? Including them would certainly change the result.

The two experiments of Cherniak et al. (2007) that were performed with natural (hydrothermally grown) quartz returned results that are similar to those obtained with synthetic quartz (see Figure below; the two data points are shown by gray triangles and are highlighted with arrows, but Cherniak et al. (2007) did not include them in their fit, which is why we do not show them in our Fig. 2.). As discussed by Jollands et al. (2020), the much

higher apparent diffusion coefficients obtained by Cherniak et al. (2007) are due to surface degradation during their experiments. Corresponding text was added to the Supplementary Information.

[REDACTED]

(b) there are also experiments from another study – Jollands et al. (2020). Do the authors see issues with this data? If not, why not use any of those data points? I'd like to know more about what makes the other experimental data unworthy of use in this case.

See above. We prefer to use TiDi01 because this experiment was performed on a magmatic quartz similar to the samples investigated in the present study. However, virtually the same temperature dependence would be obtained if the solid-source data of Jollands et al. (2020) data are used as high-T endmember.

In general, if there are issues with any of the experiments that make the authors wary of using them, that would be useful for people to know.

A statement why the experimental data of Cherniak et al. (2007) are not trustworthy was added to the Supplementary Information. An explanation why most experimental data of Audétat et al. (2021) (apart from experiment TiDi01) are too low was already given in the Supplementary Information.

Finally, what rules out the possibility of multiple diffusive mechanisms of Ti in quartz? If there are multiple mechanisms, and if there is some temperature dependence on what mechanism is operating, it seems unwise to create a diffusion relationship based on two points that are endmembers in the temperature range and utilize different methods to constrain D.

As shown above, virtually the same temperature dependence would be obtained if all the solid-source data of Jollands et al. (2020) are used as high-T endmember.

Overall, even if the low temperature diffusivities estimated in this study are correct, I am having trouble seeing how the extrapolation here to higher temperatures is appropriate. I know that everyone wants a diffusion coefficient as a function of temperature, but this seems to be oversimplifying a clearly complex problem. In light of this, I also have issues with the authors applying their equation to systems and circumstances with substantially higher temperatures than those determined from the Pinatubo samples.

See above. We think that we have sufficiently justified the validity of the temperature dependence. In addition, we like to point out that the vast majority of quartz crystals to which this diffusion chronometer is applied formed at ≤ 850 °C, which is rather close to the low-T

endmember, hence even a 1-log unit shift at 1600 °C would cause only a very minor change in the predicted diffusion coefficient at these temperatures. Corresponding text was added.

2. Magma residence times. As the authors note, the temperature at which processes are happening is especially important for diffusion chronometry, and I give them credit for considering the temperature problem thoroughly. I do still have some questions about the temperature estimates and approach (assuming the diffusivities calculated as a function of temperature are appropriate).

First, the text implies that the timescales calculated from quartz interiors are adjusted for temperature using Fig S11; however, in Table S11 it appears that only the rim times were calculated this way. The cores and interiors are instead calculated with the average D value at 710 C (i.e., -23.36 m²/s). So, this is giving a residence time at this temperature. However, the model the authors ultimately come up with for the history of the system negates this, as it implies a non-isothermal history. (e.g., line 167 and line 180-181). Evidence of resorption in many crystals also implies that the temperature got high enough at times to resorb some of the quartz. Thus, assuming a single temperature of 710 for all of the interior boundaries seems inappropriate.

We carefully checked both the main text and the supplementary information, but could not find any statement suggesting that the timescales calculated from quartz interiors were adjusted for the T vs. T_i relation in Fig. S11 (now Fig. 4 in the main text). We have modified the main text and the Supplementary Information to make more clear that with "relatively constant temperatures" we mean temperatures that fluctuated between 680 °C and 740 °C, and that the lengths of the diffusion profiles near the zircon inclusions are shaped by the *time-integrated average temperature since zircon entrapment* rather than the specific temperatures that prevailed at the time of zircon entrapment. Therefore, the use of a common, average temperature for the derivation of D-values from diffusion profiles in the quartz interiors is correct. The situation is different for the diffusion profiles measured at the contact of Ti-rich rims, as in this case the Ti-rich rims are the last events recorded in the quartz crystals. Hence, if different rims formed at different temperatures, then it makes sense to treat them individually rather than using an average temperature for all of them.

Second, the applicability of zircon saturation temperatures is not clear to me. As I understand it, the logic is: There are zircon crystals and melt inclusions in the quartz, so the melt was zircon saturated when the quartz grew and the zircon saturation temperatures from the melt inclusions reflect the temperature that the quartz grew at. A similar approach was used in a previous study (Audetat 2013) to estimate the temperature of melt inclusion entrapment. In contrast to the current study, however, the 2013 study noted that there was a large abundance of zircon crystals riddling the Bandelier quartz crystals, suggesting the system was zircon saturated throughout the growth history of the quartz. Furthermore, the quartz Ti contents were taken from the same zone where the melt inclusions were located – is this what was done in the current study as well? From the images in the appendix, that doesn't always seem to be the case (e.g., MI3 in 8b-11, MI2 in 8Db-4), in which case the correlation between T_{zirc} and T_i in quartz doesn't seem compelling.

Zircon inclusions are very common also in the samples investigated in the present study, but this it is not evident from the CL images because only exposed inclusions can be seen on this kind of images. Due to the small size of the zircon inclusions, the chance of having several zircon inclusions exposed on the same surface is extremely small. Text was added to the main text to state that "numerous zircon inclusions are present in all investigated samples", and images of a quartz phenocryst from Mt. Pinatubo (copied below) were added to the Supplementary Information to demonstrate that zircon inclusions are very abundant in these samples.

With regard to the spatial correlation between analyzed melt inclusions (→ T_{zirc}) and analyzed quartz, we are thankful that the reviewer raised this point, as we indeed forgot to show the quartz analyses that were performed next to the melt inclusions. The analyses were listed in supplementary Table S7 (now supplementary data 6), but we forgot to include them in the CL images. This has been corrected now.

In the case where the melt inclusion is located in the same zone as the zircon (and ideally the boundary used for diffusion chronometry), the approach is more convincing. But, as noted above, in most of the images in the supplementary materials, it is not clear that the zones where the zircons are located are at all related to the zones the MIs are in. So, the zircon saturation temperatures from those MIs do not necessarily reflect the temperature at which the zircon inclusions crystallized, let alone the quartz in other zones.

See above. It is the *time-integrated average temperature since zircon entrapment* that determines the length of the diffusion profiles next to the zircon inclusions, not the temperatures at the time of zircon entrapment.

Furthermore, the existence of zircon in the quartz does not mean that the quartz grew at the zircon saturation temperature – it just means that the melt was zircon saturated at some point before the quartz crystallized around it. Given the resorption and the dearth of zircon crystals in these crystals (at least compared with the Bandelier samples in the 2013 paper), is it also

possible that the system fluctuated between zircon saturated and undersaturated over the history of the system? It seems very unlikely that some melts were trapped at zircon undersaturation. First, zircon inclusions are very abundant in these samples (see above). Second, zircon saturation is usually attained at considerably higher temperatures than the ones that prevailed in our samples. Third, zircon dissolution is very sluggish, hence it takes a lot of time to re-dissolve already existing zircon grains. Based on the latter point one could actually argue that the zircon saturation temperatures obtained from rim-hosted melt inclusions represent only minimum values, as not enough time may have been available to re-dissolve an appropriate amount of zircon after the heating event. However, the good correlation of melt inclusion zircon saturation temperatures with the Ti content of the host quartz (Fig. 4, copied below) suggests that the zircon saturation temperatures are representative.

(It might be worth considering too that the zircon crystals themselves had to grow – taking the maximum size of your crystals (40 µm diameter) and the fastest existing estimates for zircon growth rates (10^{-15} m/s) would imply that these crystals had to grow for nearly a millennium before being captured in the quartz crystal – how much could temperature have changed in that time?).

Because it is the *average* storage temperature that matters for interpreting diffusion profiles in the quartz interiors, it is also irrelevant whether there was any temperature change between zircon growth and zircon entrapment.

However, one could use this zircon growth time estimate to determine the maximum error introduced by the assumption that the zircons formed immediately before entrapment. If the zircon inclusions grew 1000 years before their entrapment, then it would shift the data point of the youngest zircon inclusion by one symbol size to the left in Fig. 2a, and by half a symbol size up in Fig. 2b, whereas all other data points would remain basically the same. Hence the effect on the calculated average $\log_{10}D$ value is negligible even for this extreme endmember scenario. In reality, most of the analyzed zircons are smaller, and it is only the innermost parts of the zircons that can have grown up to 1000 years earlier, whereas the determined U-Th ages represent the integrated age of the entire grains. For these reasons we did not include this discussion.

Also, as noted in my detailed comment about Fig 3, it is interesting that the zircon inclusions all seem to have ages that are within a constrained range of time, both relative to zircon microphenocrysts and the history of the system as a whole (i.e., there are older and younger microphenocrysts, and there are no zircon grains – inclusions or microphenocrysts – younger than the Sacobia eruption). What, if anything, does this mean about the zircon inclusions being representative of the time when the quartz grew?

Regarding the first part, the apparently more constrained age range of the zircon inclusions could be accidental. One data point was previously not shown because its upper age uncertainty was "infinity", and two data points cannot be shown because already the average age plots above the infinity line in supplementary Fig. 1 (copied below). Hence, in terms of upper age limit of the zircon inclusions there is certainly no difference to the zircon

microphenocrysts. Also in terms of the lower age limit there is not much difference, except that less points are available for the zircon inclusions, but this could be accidental. Figure 3 (now Fig. 6) has been revised by adding the one previously missing zircon inclusion age, and by adding error bars to all U-Th age data.

With regard to the question why there are no zircon inclusions younger than the Sacobia eruption, it is simply because there was not much chance to find such inclusions, as they would have to be within the relatively thin, outermost growth zones. Therefore, there is no discrepancy between zircon ages and quartz growth indicated by the lack of zircon inclusion ages younger than the Sacobia eruption. A corresponding sentence was added to the Supplementary Information.

This is all to say that I am not sure that zircon saturation temperatures are necessarily providing useful information for determining the temperature of quartz crystallization related to the boundaries used for diffusion chronometry in the current study. The fact that the authors see such a wide range of T_{zirc} – up to 20 °C at 70-80 ppm Ti in quartz (Fig S11) – at a given Ti in quartz value also makes the use of this correlation for T_{qtz} somewhat unsatisfying. Again, my concerns here may just reflect the fact that I don't have as much experience as the authors with the intricacies of this dataset.

Zircon saturation temperatures obtained from melt inclusions within quartz interiors range from 690 °C to 730 °C (new Fig. 4; shown above). Sure, the correlation in this graph is not perfect, but it is also not bad ($R^2 = 0.78$), so we feel that the reviewer is complaining here at a rather high level. If the correlation equation in Fig. 4 is applied to all quartz analyses performed on the Mt. Pinatubo samples, a quartz crystallization range of 680-745 °C is obtained. This range fits very well with the range of 680-760 °C obtained from experimental phase constraints and Fe-Ti-oxides, as shown by the new Fig. 3 in the main text.

I'm also curious about the calculation of the zircon saturation temperatures - it appears that the W&H zircon saturation temperatures were selected because of their similarity to Fe-Ti oxide temperatures and a_{TiO_2} ; however, as the authors note, the Fe-Ti oxides can be unreliable, and processes like eruptive decompression can negate their robustness (even if they pass the Bacon & Hirschmann test – see your ref Hau 2021). The T_{zirc} results from the Shao 2020 model 2 are significantly higher (but also within the range of the Fe-Ti oxide temperatures) – are these being discounted based on magma composition? (I realize that the Shao et al. (2020) study suggests Model 1 is better for metaluminous to peraluminous systems, but in this study the fact that they give temperatures at the solidus makes it seem less reliable.)

The two models of Shao et al. (2020) are based on fits to all available experimental zircon solubility data, which cover a very large range of temperatures and melt compositions. As a consequence, the fit obtained for a very restricted range such as the melts of the present study is likely less good than the fit of a study (W&H 1983) that focused exactly on this kind of melt compositions and on relatively low temperatures. In the meantime, a new study on zircon solubility in silicate melts came out (Crisp and Berry, 2022). Although their solubility equation is associated with an error of ca. ± 40 °C at 700-750 °C, the results match quite well (within 20 °C) with the model of Watson and Harrison (1983) (see new Fig. 3 above).

Third, the existence of compelling resorption textures suggests that the temperature was high enough to resorb quartz at one or more times in the history of the quartz crystals (as you note, for example, in grains 8Aa-8 and 8Aa-15). Even if you do not have timescales and temperatures from these boundaries, it seems hard to interpret these textures as not reflecting some kind of resorption event. Perhaps there are other changes to the system other than temperature, but you at least rule these out for many of the boundaries. So, assuming that all of these boundaries are related to thermal fluctuations (arguable if that's fair), it is evidence of multiple temperature fluctuations. How, then, can the timescales calculated for the interior boundaries not be maxima? Even if the system cooled down and began crystallizing quartz again after a high temperature excursion, the time spent at high temperature would impact the diffusivity. Unfortunately, there is no way that I can see here that you could constrain how long the system was at these higher temperatures, or even what those temperatures were, so quantifying how much diffusion would have taken place at the higher temperatures is challenging. But, it seems important to think about.

We fully agree that most quartz crystals record several resorption events, which is why we distinguished in Fig. 3 (now Fig. 6) between growth zone boundaries associated with strong resorption, and ones without evident resorption.

Thermal models of magma chamber rejuvenation consistently predict that temperatures first increase very rapidly and then immediately decrease rapidly again, followed by an asymptotic convergence to new equilibrium conditions (e.g., Bachmann and Huber, 2016; Rubin et al., 2017; Kent and Cooper, 2018).

[REDACTED]

The Figure above is from Rubin et al. (2017)

[REDACTED]

The Figure above is from Bachmann and Huber (2016)

[REDACTED]

The Figure above is from Kent and Cooper (2018)

In contrast to zircon, quartz can respond to temperature changes relatively quickly. This means that during the heating events quartz dissolved, whereas during the immediately following cooling periods it precipitated again. From the above it follows that (1) the time intervals that are not recorded in the quartz *due to heating* should be comparatively short (in contrast, there can be very long time intervals without record due to storage at constant temperature), and (2) due to the rapid re-precipitation of quartz during cooling, the maximum temperature that was reached during a heating event should be closely approximated by the most Ti-rich quartz of a given CL-bright growth zone, which should in principle be the oldest/innermost part of that growth zone. Indeed, in many growth zones the most CL-bright

quartz was precipitated first, followed by precipitation of gradually less CL-bright quartz (see example below taken from Appendix B):

QtzIIb-10

In summary, we see no risk that the quartz crystals equilibrated at significantly higher temperatures than those constrained by the maximum quartz Ti contents, and the time spans that were spent at these maximum temperatures should have been relatively short compared to those spent at lower temperatures. A corresponding discussion was added to the main text.

A few additional comments and questions:

Line 156: missing end parenthesis
corrected

Line 174-176: the implication in this sentence is that the system went through multiple periods of recharge. How did the system vacillate between warm and cold but not produce evidence of resorption? **Good point. The text was changed to "one or several magma recharge event(s)".** Maybe they were relatively short-lived events or didn't reach temperatures where the quartz was resorbing, but this sentence at least implies that there were temperature fluctuations that would impact the diffusion coefficients.

Yes, but as discussed in the main text and in the Supplementary Information, the last heating event that led to quartz resorption and subsequent growth of Ti-rich quartz rims was apparently too short-lived to significantly affect the lengths of the diffusion profiles within the quartz interiors, otherwise the diffusion coefficients would not correlate with the zircon ages, and there would be no gradual blurring of diffusion profiles towards crystal centers. Due to the importance of the latter observation, corresponding CL images were added to revised Fig. 1.

Figure 3: Using the same color for "unclear" and "no resorption" in this plot is a bit misleading. Looking at the images for some of the grains, (e.g., 8Da-1), the contact where the interior diffusion profiles have been taken does not at all appear to be normal, and that deserves delineation on this plot.

We gave our best to classify the contacts in an objective, neutral way. Grain 8Da-1 (reproduced below) is complicated. It apparently consists of three individual crystals that grew into each other: one crystal at the upper right (notice the arcuate boundary to the right of point 67), one crystal at the bottom right (notice the arcuate boundary above points 53 and 45), and the main, remaining crystal. This is why the CL texture in this grain looks so abnormal.

8Da-1

Within the three individual sub-crystals, including the one at the bottom right in which the diffusion profiles were measured, it is hard to say whether or not there was quartz dissolution prior to the formation of the CL-bright quartz in the crystal interior. This is why we classified the two boundaries with the $\log_{10}Dt$ values of -10.58 and -10.65 as "unclear / no resorption".

How do you explain the younger "other" contact in 8Aa-8? We are not sure which contact the reviewer refers to. The one with $\log_{10}Dt = -11.40 \text{ m}^2$? (for convenience, the CL image of 8Aa-8, which is now shown in revised Fig. 1a, is copied below). If yes, then it points to a heating event that took place about 23,000 years ago, i.e., at about the time of the Sacobia eruption. This quartz crystal could thus stem from an area of the crystal mush that did not erupt at that time.

Why are all the of the zircon inclusions from a fairly tightly constrained age range? See above; the overall range is actually the same, and the tighter clustering at 100,000-200,000 years could be coincidental. Geochemically the included zircons look significantly different from the microphenocrysts as well. Is this related to a size bias, since you are trying to select the smallest included zircon crystals? Or does it mean something about the other crystals (e.g., older grains are xenocrysts)? In the Supplementary Information we provide a lengthy discussion about this geochemical difference. It is partly due to the more evolved nature of the melts from which the zircon inclusions grew (in order to reach quartz saturation, at least 60% of the dacitic melt had to be crystallized), partly due to changes in the zircon/melt partition coefficient during melt evolution, and partly due to sampling bias of the zircon microphenocrysts. Similarly, why are there so few younger grains? In general, why are all the analyzed zircons seemingly Inanaro to a bit younger in age? See above. Zircons younger or equal than the Sacobia eruption would be found almost only in the thin rims. What does this mean about your assumptions of zircon saturation? See responses above. If you compare similarly sized microphenocrysts and inclusions, do you estimate similar ages and/or temperatures? Yes, if similar-sized zircon microphenocrysts could be extracted analyzed, we would expect similar characteristics as the zircon inclusions in quartz. However, all analyzed microphenocrysts are orders of magnitude larger.

Table 1: The acronyms for the country should be in the caption Done

Lines 442-446: Why were the standards not reproducible? The analysis of trace elements in quartz by electron microprobe is challenging. The reason why on that particular day the measurements on the quartz standard were slightly off (but for Ti within error still in agreement with the published reference value) is not known. The problem seems to be well solved with the applied correction, as after that correction the concentrations obtained by EPMA match well with those obtained by LA-ICP-MS, as shown by supplementary Fig. 7.

Line 463: what does 'mostly glassy' mean? Were they devitrified or is it just that they sometimes also contained a bubble and crystals? The "mostly" was deleted.

Reviewer #2 (Remarks to the Author):

In this manuscript, the authors use Ti diffusion profiles in natural quartz to extract diffusion coefficients, with absolute time scales constrained using zircon ages. The concept is novel and solid, and the data are high quality. There is a major issue that has not been addressed satisfactorily, which is the nature of the initial condition prior to Ti diffusion. The inherent assumption in the model is that the Ti profile

is a perfect step function, but it is not clear how this assumption is backed up by the data/textural observations etc. What is the evidence that growth is not accompanied by some progressive change in the composition of the growth medium, leading to a gradual change in the quartz composition? Then, the initial condition is not a step function, and diffusion only smooths out some pre-existing gradient. The result would then be that the diffusion coefficients determined are maxima.

This important point has been raised also by reviewer 3. For the zircon-related diffusion profiles (and, hence, for the estimation of the diffusion coefficient at 710 ± 30 °C) we can clearly rule out a significant effect due to deviations from ideal step functions, based on the following reasoning:

(1) Many CL-bright growth zones are preceded by visible resorption events, as evident from the discordant cutting of older growth zoning and from the rounded shape of the truncation surface. In these cases, the original steps can be expected to have been relatively sharp – at least as sharp as those that are now present at the contact of the outermost growth zones. Even if other CL-bright growth zones show no visible textural evidence for prior resorption, it is well possible that small amounts of quartz were dissolved prior to the CL-bright quartz precipitation also in many of these cases. In the revised manuscript we have made sure that only profiles measured at the inner contact of CL-bright growth zones are considered.

(2) In many crystals the degree of blurring increases very consistently from rim to core (see e.g. Fig. 1b, c in the revised Fig. 1 above). This simply cannot be a coincidence and therefore suggests that all contacts were initially relatively sharp – probably at least as sharp as the current contacts of the outermost growth zones, which give $\log_{10}Dt$ values ranging from -13 to -14. For the $\log_{10}Dt$ values determined next to zircon inclusions (and thus for the calculated diffusion coefficients in Fig. 1C) it makes virtually no difference whether an initially infinitely sharp step function is assumed or one that corresponds to a $\log_{10}Dt$ value of -13 to -14, as even for the shortest zircon-related diffusion profile with a $\log_{10}Dt$ value of -11.94 this changes the value only on the second decimal place. Therefore, we are confident that the $\log_{10}Dt$ values associated with zircon inclusions are real and were not significantly affected by deviations from the assumption of initially sharp step functions.

Corresponding text and Figures were added to the main text.

The issue of the initial diffusion boundary conditions become more relevant for the residence times retrieved from the quartz rim contacts, as for these time it indeed matters whether the initial Ti distribution was a perfect step function, or whether some degree of blurring was present already at the beginning. However, if the diffusion profiles at the contacts to the quartz rims were dominantly reflecting pre-existing Ti concentration gradients, why should there be a positive correlation with erupted magma volume (revised Fig. 7)? This observation suggests that the calculated residence times are meaningful, as there is no reason why quartz crystals in large eruptions should have had consistently wider initial Ti concentration gradients than those in small eruptions. Therefore, even if we cannot exclude significant contributions from pre-existing gradients, we believe that the diffusion profiles at quartz rim contacts still reflect dominantly residence times. Corresponding text was added to the Supplementary Information.

As a result, the new data point determined from natural quartz is not a single point on an Arrhenius plot, and the extrapolation between an experimental datum and this one is not universally applicable. This issue then propagates into all of the other time scales. Therefore, whilst this study provides high quality data, I would not recommend publication in this form. As it stands, the manuscript will further confuse the Ti in quartz debate. The authors should either provide clear and solid evidence that the initial condition prior to diffusion in the natural system was a step function, or alternatively reframe the discussion around the new data simply providing a maximum diffusion coefficient for Ti, and that the actual diffusion coefficient must be somewhere below this value. This would obviously be a major revision. See above. We are confident that the calculated diffusion coefficients are not significantly affected by deviations from initial step functions, and we present evidence that even the diffusion profiles at the quartz rims are dominantly caused by Ti diffusion rather than reflecting initial gradients.

Reviewer #3 (Remarks to the Author):

Dear Dr. Andreas Audétat,

It was a pleasure to review your manuscript. As you know, I agree that Ti in quartz diffusion is one of the great problems facing petrologists who utilize the technique for magmatic time scales. I found the writing to be thoughtful and well researched, mostly. For me, there are two main flaws that crop out during the review. I think that this is an important enough issue to suggest that these two flaws should be addressed before this can be published. I would hate to continue to add controversy to this problem, when you have a real chance to provide some clarity.

Below I'll outline my two main issues, and I hope you find the comments helpful.

Best,

Dr. Joe Boro

Senior Staff Scientist and EPMA Lab Manager

Sandia National Labs, USA

1. You correlate greyscale CL to Ti concentration and use it as an exact proxy. This is incorrect to do, especially as it appears the data were collected on an SEM and not a microprobe (explained further below). The reasons to avoid this are vast, and although you provide a plethora of researchers who choose to do this (and you've done this before), they are also incorrect, so I'm afraid your application of the technique here only muddies the waters further.

We agree that a definite proof for this relation was missing (it is provided now), but we disagree with some of the reviewer's arguments.

Boro et al. 2021 (Am. Min.) as well as other authors, give a large list of why not to do this, and I can summarize here for you:

a. Boro et al. (2021) find Ti profiles measured by EPMA and overlapping greyscale CL profiles do not match in most cases, often the grey-scale CL showing a steeper or more diffuse gradient.

We had a very careful look at this paper. It turns out that in most cases the number of EPMA data points covering the diffusion profile is too low to allow any reliable conclusions to be drawn regarding the steepness of the gradient (see examples below).

[REDACTED]

Matthews et al. (2012), their Figure 7a, also show this mismatch (even though they ignore it's significance in their data interpretation).

See below. The mismatch in Matthews et al. (2012) is well explained by the different analytical setups.

b. While spectroscopic measurements of quartz CL emissions at specific wavelengths, such as ~454 nm, are very well correlated with Ti contents (MacRae et al. 2013, 2018), grayscale CL is not a reliable proxy for Ti concentration.

Yes, although we demonstrated a general correlation between CL intensity and quartz Ti content in supplementary Fig. S7, we could not rule out the possibility that the grayscale

profiles in our panchromatic CL images were also significantly affected by other trace elements or crystal defects. For this reason we bought a shortpass filter that allows only light with wavelengths <500 nm to pass through, and longpass filter that allows only light with wavelengths >500 nm to pass through, and then took CL images of the same crystal areas once in panchromatic mode (which collects all light from 200 nm to 900 nm), once with the shortpass filter placed in front of the detector, and once with the longpass filter placed in front of the detector. The reason for choosing 500 nm is that this wavelength efficiently separates Ti-related CL emissions that peak at ~450 nm (~2.8 eV) from CL emissions caused by other factors such as the non-bridging oxygen hole centers (NBOHC) that peak at ~650 nm (~1.9 eV), as shown by the two hyperspectral maps of magmatic quartz phenocrysts below.

The results revealed that in all cases (three crystals of Mt. Pinatubo; one crystal each from the other occurrences) the <500 nm (i.e., Ti-related) CL images look virtually identical to the panchromatic images, whereas the >500 nm images look completely different. Correspondingly, diffusion profiles measured in the <500 nm CL images are within uncertainty (which reflects mostly the reproducibility of the profile selection) the same as those measured in the panchromatic images. Furthermore, CL images taken from the same contact at different magnifications return the same $\log_{10}Dt$ values, demonstrating that the profiles do not change as a function of irradiation time. Hence, we with the help of these images we are now able to unambiguously demonstrate that the grayscale profiles measured in our panchromatic CL images faithfully reflect Ti concentration profiles.

Corresponding text plus the Figure shown above has been added to the Methods section in the main text, whereas all comparative CL images of all other samples are provided in Appendix A.

c. Also, as described by Leeman et al. (2012) and MacRae et al. (2018), grayscale intensities include not only CL from Ti dopants but also from the intrinsic CL produced by quartz, by crystallographic defects, by aluminum dopants, and by non-bridging oxygen holes in the crystal structure.

See above. With the 500 nm shortpass-filter we could efficiently block the signals produced by these other CL-emitters.

d. For many panchromatic CL detectors, the collection optics accept light from a wide variety of angles, and therefore light from adjacent areas, which are still emitting from the initial electron bombardment, or light transmitted through the sample and scattered off microcracks in the sample, which may be included in a CL measurement that is supposed to represent only the CL emitted from an individual pixel (MacRae et al. 2013). See above. The profiles are stable under the electron beam, and the spatial resolution that can be achieved with our analytical settings has been demonstrated in supplementary Fig. S1 (now Fig. 8).

e. This should be especially avoided in SEMs where there are not optical focusing capabilities. Often times stray photons can bounce off the interior of the chamber and add noise to measurements, artificially adding counts to adjacent pixels during the raster or line scan, and creating artificially diffuse profiles. See above.

i. One exercise you could do, is to use extremely low kV (2-5 kV) with fast scan rates, to help with any over excitement of the bonds. See above.

f. This issue isn't as much of a problem outside of the growth boundaries and is likely why you see CL correlating with Ti in the spot analyses presented in your supplemental data. See Matthews et al., their Figure 7a especially demonstrates the mismatch at the diffusion boundary, but the correlation outside the boundary. I think this has something to do with the intrinsic defects which get included during rapid growth of the quartz, which are strong CL producers.

The mismatch in the profiles of Matthews et al. (2012) is simply due to fact that CL is a surface measurement, whereas the μ -XRF penetrates to much higher depth and the incident beam had angle of 45°C, as the authors explained on p. 1401 and illustrated in their Figure 10, which is copied below.

[REDACTED]

Boro et al. (2021) also sees good correlation outside the growth boundary. For what it's worth, later on we did collect spectral CL at 452 nm, and were able to match that pretty well with the EPMA Ti data, so if you have access to a spectral CL system, I suggest using that as it is better proven to act as a proxy for [Ti]. I realize collecting trace quant data for Ti in quartz at these scales is difficult, but methods do exist (nanoSIMS, oblique EPMA profiles, etc.), or at worst, get some spectral CL data. See above. With the help of the optical filters we are now able to demonstrate that the previously measured profiles in panchromatic CL images indeed represent Ti concentration profiles.

g. Lastly, there is no proof that Ti chemical profiles are thought to start as step functions, which is fundamental to this paper to carry any meaning. A study which examines [Ti] at T=0, say in growth experiments would be much more impactful and helpful in understanding the diffusivity discrepancies. See above. To summarize, you are assuming greyscale CL is a direct proxy for Ti, which we know isn't true It is definitely true for our samples and our analytical setup, especially at diffusion boundaries, and then using unknown T=0 diffusion gradients to extract a diffusivity coefficient. The reasons why the assumption of initial step functions is justified are given above. I hope you can see why this raises red flags in my mind.

2. You assume that the zircon inclusions found in the quartz grew and formed at the same time as the rims of the quartz. I find this assumption to also be simplified and ignoring some simple thermodynamics about zircon saturation. For example, if Ti activity is increased due to elevated temperature in a system to increase the Ti in a quartz rim, zircon wouldn't be crystalizing, and possibly resorbing. As I see in most of your images, the zircons are lacking idiomorphism and are in most cases rounded, looking to be resorbed, suggesting they existed in the system for some time prior to being incorporated in the quartz rim at a time when they themselves were not in equilibrium. Unfortunately we don't fully understand this comment. The depicted zircon inclusions certainly did not form at the same time as the quartz rims. Maybe the reviewer meant "growth zones" instead of "rims"? But why should the Ti activity increase during a temperature increase? Supplementary Table S7 (now supplementary Data 6) shows that this is not the case (the \$a_{\text{TiO}_2}\$ of the rim-hosted melt inclusions is the same as that of melt inclusions within the quartz

interior). Most zircon inclusions are in fact relatively long needles (as visible in the added transmitted-light images in supplementary Fig. 8), but for the dating we preferred the more stubby ones because they were easier to fully expose during polishing for the U-Th dating.

REVIEWER COMMENTS

Reviewer #1 (Remarks to the Author):

The authors do appear to have addressed my comments in the actual text.

Reviewer #2 (Remarks to the Author):

Review of Audetat et al

From going through the manuscript, and responses to the many reviewer comments from the first round, it looks like everything has been well addressed. The big potential issue was initial conditions, but this is addressed well.

There is one change that could be made, which in my opinion could hugely increase the usefulness of the new dataset. At the moment, the authors make the point that diffusion chronometry is hard, because there are a few orders of magnitude difference between the fastest and slowest D s for Ti in quartz. The study doesn't help this at all, because the new D is somewhere in the middle, so the big range still exists.

Instead, why not use the new $\sim 700\text{C}$ data point, which, as far as I can gather, is trustworthy, as a low temperature pin point for the experimental diffusion coefficients? From looking at Fig 5, it seems like you could refit all of the Jollands et al data with this new data point. Then, rather than just adding a fourth set of diffusion coefficients, you are 'refining' one of the published ones using this high-quality low T data. A data point at 700C would be basically impossible to get by any experimental means. Therefore, you would make a new Arrhenius equation which is better than all of the previous ones. I expect that including more data points in the regression would have basically no effect on the Arrhenius relationship that you have derived, but it would look so much better to the community if there is some consensus between natural and experimental. Choosing a single data point from an experimental campaign doesn't look good, and just adds more muddiness to the already murky Ti-in-quartz problem. Also note that low T diffusion experiments are really hard to do, so we should have least trust in any of the low T data, either those of Jollands or Audetat.

If you did such a regression, with the new data point, I would absolutely recommend this as the go-to D calibration for Ti in quartz.

The sticky issue would remain, though, which is choosing the Audetat or the Jollands calibration for the high T points.

Also, please emphasise ad nauseam that this '2nd decimal place' result is only for this study. I can imagine people citing this as a reason to not think about initial conditions at all.

Reviewer #3 (Remarks to the Author):

I believe you have sufficiently addressed many of the problems and acknowledged the possible short comings of the study.

I still believe you need to be extremely cautious when using CL as a proxy for Ti in quartz, and this is a dangerous practice which could lead others into misapplying the method. The filters you use sound like a stop gap, but ultimately you should look to use spectral CL in future studies.

RESPONSE TO REVIEWER COMMENTS

(comments shown in black; responses shown in blue)

Reviewer #1 (Remarks to the Author):

The authors do appear to have addressed my comments in the actual text.

Reviewer #2 (Remarks to the Author):

Review of Audetat et al

From going through the manuscript, and responses to the many reviewer comments from the first round, it looks like everything has been well addressed. The big potential issue was initial conditions, but this is addressed well.

There is one change that could be made, which in my opinion could hugely increase the usefulness of the new dataset. At the moment, the authors make the point that diffusion chronometry is hard, because there are a few orders of magnitude difference between the fastest and slowest D s for Ti in quartz. The study doesn't help this at all, because the new D is somewhere in the middle, so the big range still exists.

Instead, why not use the new ~700C data point, which, as far as I can gather, is trustworthy, as a low temperature pin point for the experimental diffusion coefficients? From looking at Fig 5, it seems like you could refit all of the Jollands et al data with this new data point. Then, rather than just adding a fourth set of diffusion coefficients, you are 'refining' one of the published ones using this high-quality low T data. A data point at 700C would be basically impossible to get by any experimental means.

Therefore, you would make a new Arrhenius equation which is better than all of the previous ones. I expect that including more data points in the regression would have basically no effect on the Arrhenius relationship that you have derived, but it would look so much better to the community if there is some consensus between natural and experimental. Choosing a single data point from an experimental campaign doesn't look good, and just adds more muddiness to the already murky Ti-in-quartz problem.

Also note that low T diffusion experiments are really hard to do, so we should have least trust in any of the low T data, either those of Jollands or Audetat.

If you did such a regression, with the new data point, I would absolutely recommend this as the go-to D calibration for Ti in quartz.

The sticky issue would remain, though, which is choosing the Audetat or the Jollands calibration for the high T points.

We have changed the fitting approach as suggested by the reviewer, using the experimental data of Jollands et al. (2020) for high temperatures, and forcing the regression through the new low-T data point obtained from Mt. Pinatubo. Explanations why we prefer the experimental data of Jollands et al. (2020) over the ones from Aud  tat et al. (2021) are now given in both the main text and in the Supplementary Information. The temperatures calculated with the new fitting equation are so similar to the previous ones that Figures 6 and 7 remain unchanged.

Also, please emphasise ad nauseam that this '2nd decimal place' result is only for this study. I can imagine people citing this as a reason to not think about initial conditions at all.

We have changed the sentence to "...as **in this specific case** the extracted $\log_{10}Dt$ values are affected only on the second decimal place."

Reviewer #3 (Remarks to the Author):

I believe you have sufficiently addressed many of the problems and acknowledged the possible short comings of the study.

I still believe you need to be extremely cautious when using CL as a proxy for Ti in quartz, and this is a dangerous practice which could lead others into misapplying the method. The filters you use sound like a stop gap, but ultimately you should look to use spectral CL in future studies.